# PHYSGYM: Benchmarking LLMs in Interactive Physics Discovery with Controlled Priors

Yimeng Chen[1]*, Piotr Piękos[1], Mateusz Ostaszewski[1], Firas Laakom[1], Jürgen Schmidhuber[1,2,3]

[1]Center of Excellence for Generative AI, KAUST
[2]The Swiss AI Lab, IDSIA-USI/SUPSI [3]NNAISENSE

⊙ PRINCIPIA-AI/PHYSGYM

## Abstract

Evaluating the scientific discovery capabilities of large language model based agents, particularly how they cope with varying environmental complexity and utilize prior knowledge, requires specialized benchmarks currently lacking in the landscape. To address this gap, we introduce PHYSGYM, a novel benchmark suite and simulation platform for rigorously assessing LLM-based scientific reasoning in interactive physics environments. PHYSGYM's primary contribution lies in its sophisticated control over the level of prior knowledge provided to the agent. This allows researchers to dissect agent performance along axes including the complexity of the problem and the prior knowledge levels. The benchmark comprises a suite of interactive simulations, where agents must actively probe environments, gather data sequentially under constraints and formulate hypotheses about underlying physical laws. PHYSGYM provides standardized evaluation protocols and metrics for assessing hypothesis accuracy and model fidelity. We demonstrate the benchmark's utility by presenting results from baseline LLMs, showcasing its ability to differentiate capabilities based on varying priors and task complexity.

## 1 Introduction

Automating aspects of the scientific discovery process holds immense promise for accelerating research across physics, chemistry, and biology [1, 2, 3, 4, 5, 6]. A central capability for artificial intelligence (AI) agents in this pursuit is scientific reasoning, i.e. the ability to explore environments, gather evidence, form hypotheses and uncover underlying mechanisms [3, 7, 8, 9]. The idea of an autonomous "AI Scientist" dates back at least more than three decades to early work on artificial curiosity [10, 11, 12], which proposed agents that learn to explore in order to improve their internal world models. Recent advances in Large Language Models (LLMs) [13, 14], trained on a vast corpus of scientific literature and equations, have sparked enthusiasm about their potential as general-purpose scientific agents [14, 9, 15, 16]. Systems like the *AI Scientist* [9, 17] demonstrate closed-loop scientific workflows: generating hypotheses, designing experiments, analyzing results, and even writing papers. Other systems like SCIMON [18] and SCIAGENTS [19] leverage literature-based reasoning and multi-agent coordination to discover hypotheses and explore novel research directions. Yet despite these advances, a fundamental question remains: *how do these models actually reason in scientific discovery?*

The answer is obscured by a critical limitation in how we evaluate these models. Existing evaluation frameworks [20, 21, 22, 23, 24] rely on static datasets or expose fixed sets of priors, lacking fine-grained control over what contextual knowledge is available to the model. However, scientific reasoning depends critically on context. Consider a simple pendulum experiment: if an agent is told

---

*Correspondence to: `yimeng.chen@kaust.edu.sa`

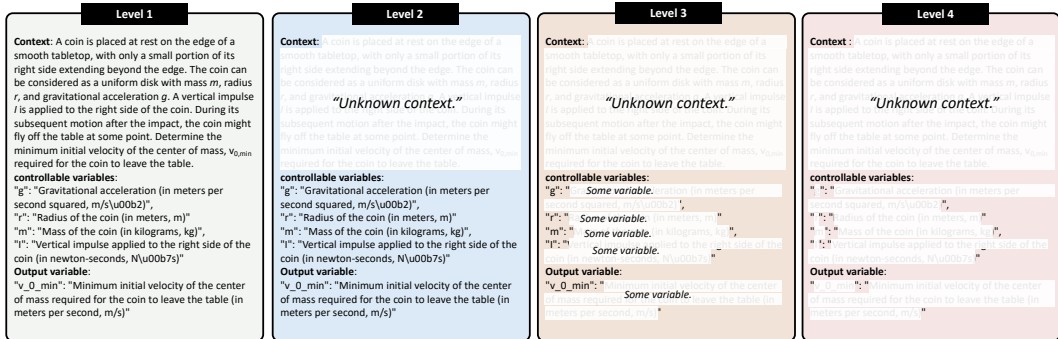

Figure 1: Controlled levels of prior knowledge. PHYSGYM lists three types of prior knowledge: *Context* - a textual description of the environment, *Variable Descriptions* and *Variable Names*. The levels of prior start with full information disclosed to the model at *Level 1* and then gradually strip the information from the model - removing the *Context* at *Level 2*, *Variable Descriptions* at *Level 3* and then finally removing *Variable Names* as well by anonymizing variables at *Level 4*.

that the system is a harmonic oscillator and sees variables like "length" or "gravity" it can trivially match patterns to recover the canonical solution. However if those variables are anonymized or the description is hidden, the agent must engage in experimental probing. It has to vary inputs, observe outputs, and form structural hypotheses to discover the relationships. These scenarios involve identical physics, but radically different cognitive demands. This distinction between memorization and true mechanistic inference lies at the heart of scientific reasoning, yet current benchmarks fail to disentangle them. This motivates the need for interactive, *controllable* benchmarks that evaluate AI models on how they adapt to unfamiliar problem settings, balance prior knowledge with posterior exploration (leveraging existing priors or compensating for missing ones through experimentation), and their ability to construct and modify physics models.

To address this gap, we introduce PHYSGYM, a novel benchmark suite and simulation platform designed for the assessment of LLM-based scientific reasoning within interactive physics environments with *controllable priors*. At its core, physics discovery can be abstracted as a curiosity-driven search for action sequences (experiments) that generate data containing a previously unknown yet learnable regularity or compressibility [8], eventually leading to minimal descriptions of the data that explain the observations [25, 26, 27, 28, 29]. PHYSGYM operationalizes this abstract process within a series of carefully designed, interactive physics environments. In each environment, an agent's task is to discover the equation relating a target observation to a set of controllable variables. To achieve this, agents iteratively design experiments by proposing values for the controllable variables and receiving the corresponding outcomes as feedback.

A central feature of PHYSGYM is its fine-grained control over the prior information available to the agent. The platform provides structured environmental descriptions that detail the experimental setup, the physical meaning of variables, and their symbolic representations. By selectively revealing or masking these information (as illustrated in Figure 1), we can systematically investigate how varying degrees of prior knowledge affect an agent's problem-solving and reasoning capabilities.

Through a comprehensive interface, agents in PHYSGYM interact with the environment, control experiments, and systematically document their discovery process. This interface also enforces realistic constraints that mirror actual scientific practice, most notably a limited experimental budget. To assess performance, PHYSGYM measures both the fidelity between a discovered hypothesis and the ground-truth physics equation, as well as how well the hypothesis fits the observed data. Together, these components provide robust insights into an agent's scientific reasoning capabilities under realistic conditions.

To demonstrate the utility of PHYSGYM and establish initial performance levels, we present baseline results using representative LLMs with direct prompting. Our evaluation across four prior knowledge levels reveals that while aggregate success rates generally decrease with reduced priors (e.g., from 66% to 31% for top-performing models), individual problems exhibit non-monotonic patterns where some are solved only at lower prior levels but fail at higher ones. Through detailed case studies, we identify specific failure modes including inadequate exploration strategies, overreliance on

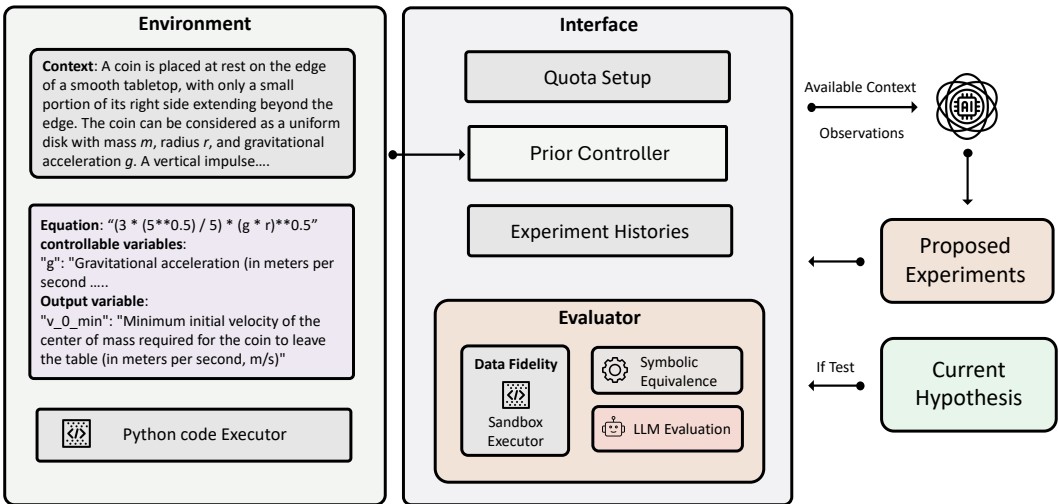

Figure 2: Overview of the PHYSGYM suite.

memorized patterns, and inability to reason about causal relationships between variables. These findings demonstrate that PHYSGYM provides valuable insights into model reasoning strategies and highlight concrete areas for developing more robust AI agents for scientific discovery.

In summary, this paper makes the following contributions:

- PHYSGYM: A novel benchmark suite and simulation platform for evaluating interactive LLM-based scientific reasoning, designed for systematic control over task complexity and prior knowledge.
- Tailored evaluation protocols and metrics for assessing performance and reasoning processes within the PHYSGYM framework.
- Experimental validation of PHYSGYM using representative LLMs, demonstrating its effectiveness in differentiating their capabilities and revealing novel phenomena including the conflict between knowledge recall and discovery through detailed case studies.

## 2 Related Works

**AI Scientists.** Early research in computational scientific discovery conceptualized discovery as a heuristic search process, predicated on the view that creativity is an emergent property of specific computational mechanisms rather than an ineffable quality [30, 3, 31]. Systems from this era, such as BACON [1] and STAHL [32], demonstrated the capacity to rediscover known scientific laws from structured data, while others like AM [33] and Eurisko [34] applied similar heuristic methods to explore mathematical concepts. Later, Schmidhuber [5, 35, 8] established a formal, computable theory of artificial curiosity and creativity [10, 11, 12] grounded in intrinsic motivation. This framework views inductive science as an active process of data collection and compression, where an agent actively generates experiments resulting in data that contains previously unknown patterns that allow for better compressing the observations through shorter programs. The architecture involves a reinforcement learning-based controller that selects actions to generate new data for an adaptive world model which, in turn, learns to predict or compress the agent's sensory history. The intrinsic reward that drives the controller is the measured learning progress of the world model as its compression of the perceptual data improves.

Recent advances in large language models (LLMs) have revived these visions, producing autonomous scientific agents that can generate hypotheses, design and run experiments, and iterate on findings. Systems like the AI SCIENTIST [9] and its successor [17] demonstrate closed-loop scientific workflows: generating hypotheses, designing experiments, analyzing results, and even writing papers. Other platforms like SciAgents [19] and SciMuse [36] coordinate multiple agents or leverage massive knowledge graphs to uncover non-obvious research directions and interdisciplinary insights. The ability to explore and refine ideas has been recognized as a central ability of AI scientists. For instance, SciMON [18] optimizes hypotheses for originality using literature-grounded novelty scoring, while

ResearchAgent [37] simulates peer-review loops to refine research ideas iteratively. Ma et al. [38] proposed a bilevel optimization-based framework, where the outer-level leverages LLMs to conduct reasoning and generate hypotheses, while the inner-level employs simulations to provide feedback and perform numerical optimization. Emerging work also explores how agents can generate or evolve themselves: ADAS [39] evolves agent designs via meta-level optimization, while OMNI-EPIC [40] proposes tasks with increasing "interestingness", forming a self-curated curriculum for exploration.

Designing benchmarks to evaluate AI capabilities in scientific research has become crucial for inspiring and advancing the development of AI scientists. Existing research has proposed several benchmarks to assess AI performance in scientific tasks. Current benchmarks primarily focus on coding [41, 42], problem-solving [43], and research engineering abilities within specific scientific domains for predefined tasks and datasets [44, 45], including training models, preparing datasets, and conducting experiments. However, benchmarks for fundamental research capabilities, such as experimental design and the formulation of reasonable hypotheses, remain largely underexplored.

**Equation discovery.** Scientific inquiry often begins with the classification of objects and the construction of taxonomies, but as a field matures, it increasingly emphasizes quantitative modeling through mathematical equations. This progression has led to the formalization of equation discovery—the task of uncovering underlying mathematical relationships from data or interactive environments—as a key research direction [1, 3, 46]. Closely related is the field of symbolic regression, which aims to infer interpretable mathematical expressions from data [47, 48, 49].

LLMs have recently emerged as a promising paradigm for scientific equation discovery, offering new capabilities grounded in their broad scientific pretraining [50, 23, 51, 9]. Several LLM based methods have been developed for equation discovery. For instance, In-Context Symbolic Regression [52] leverages LLMs in an iterative error-correction loop, yielding simpler expressions with enhanced generalization capabilities. LLM-SR [50] frames the task as program synthesis, combining LLM-generated symbolic skeletons with evolutionary search and parameter optimization to outperform traditional methods across multiple scientific domains. LLM4ED [53] guides equation generation and optimization through natural language prompts, employing self-improvement and evolutionary strategies to uncover governing equations in nonlinear dynamical systems. LLM-Feynman [54] integrates LLMs with Monte Carlo tree search and self-evaluation, showing the ability to rediscover over 90% of the Feynman physics equations. However, as current approaches are tailored to static datasets [22], the ability of models to generate hypotheses and design experiments remains inadequately assessed.

**Interactive benchmarks.** In order to assess the full capabilities of different agents in a complete scientific discovery process, there has been growing interest in developing interactive benchmarks that go beyond static datasets. These environments aim to capture the dynamic nature of scientific inquiry, where agents must iteratively design experiments, collect data, and revise hypotheses based on feedback. Science-Gym [20] provides 5 simple physics simulation environments where agents must achieve desired states by modifying the parameters of physical objects. However, as it was not designed for large language models, it does not consider the novelty of physics problems that presents unique challenges for LLM evaluation. CodeARC [55] focuses on inductive program synthesis discovery. Discovery World [21] offers 24 simulation environments structured across 8 distinct domains and 3 difficulty levels. Agents are required to perform various actions within these environments to accomplish specified task objectives. The environmental interactions typically involve common actions such as taking or moving objects. In contrast, PHYSGYM leverages 97 novel physics problem settings spanning 6 distinct physics domains. Our framework establishes a more abstract and simplified environment with enhanced specificity, while providing fine-grained control over the prior knowledge embedded in linguistic descriptions—a core differentiating factor when evaluating LLMs compared to other models. This design enables more targeted assessment of language models' scientific reasoning capabilities within controlled experimental conditions.

## 3 The PHYSGYM Benchmark

A key process in scientific discovery consists of formulating hypotheses based on prior knowledge and observational frameworks, carrying out experimental validation, and accordingly refining and developing theoretical models. PHYSGYM is designed to simulate this process in a controllable abstract setting.

Specifically, we simulate an *environment* whose characteristics are described textually and which encompasses both controllable and observable *variables*. The latent relationship among these variables is governed by a fixed, albeit unknown, mechanism function $f$. The task of the AI physicist is to deduce a *hypothesis* $\hat{f}$ that is consistent with the true underlying mechanism of this environment. This must be accomplished based on the available information and conducting *experiments* within the environment, subject to a limited budget. An experiment consists of a specific assignment of values $\mathbf{x} = (x_1, \ldots, x_n)$, to controllable variables $\{x_i\}_{i=1}^n$. Subsequently, the AI physicist can observe the environmental outcomes, i.e., the corresponding values of the observable results $f(\mathbf{x})$.

This section introduces the proposed benchmark. We detail the structure of the dataset, the accompanying simulation environment designed to facilitate experimentation, and the evaluation metrics.

## 3.1 Dataset Construction

Our current dataset is constructed from 97 physics problems selected from PHYBench [56]. These problems feature comprehensive descriptions of their physical context, and their solutions are presented as symbolic equations that define the relationship between a target physical quantity and other relevant physical quantities. Furthermore, a complete derivation process, grounded in physics principles, is provided for each problem. Consequently, this collection offers a robust foundation, facilitating the development of our abstract environment.

The construction of the data for each problem was a meticulous process involving the assistance of LLMs followed by rigorous manual verification and refinement by domain experts. This hybrid approach allowed us to efficiently process the initial information while ensuring accuracy and relevance. Each data instance in our benchmark is structured with the following fields:

- `context`: This field contains the textual description of the physics problem. It serves as the contextual or environmental description from which a model must derive understanding.
- `solution`: A manually provided, step-by-step reasoning process that solves the problem based on established physics principles. This offers a gold-standard derivation.
- `equation`: This represents the core physical relationship of the problem, expressed as an equation linking input and output physical quantities. It acts as the ground-truth model of the environment.
- `python_code`: An executable Python script that implements the `equation`. This code can simulate the environment's behavior by taking numerical inputs and producing the corresponding output. Crucially, it also includes checks for the numerical validity of inputs and outputs.
- `input_variables`: A list of variables that serve as inputs to the `equation`. Each variable is accompanied by its physical description and its corresponding unit (e.g., "mass" in "kilograms").
- `output_variable`: The output variable that the `equation` predicts. Similar to input variables, it includes a detailed physical description and its unit (e.g., "velocity" in "meters per second").
- `dummy_variables`: This field lists variables that are mentioned within the `context` (problem description) but are not causally related to the `output_variable` through the core `equation`.

## 3.2 The Simulation Environment

Beyond the dataset itself, PHYSGYM includes a comprehensive simulation environment designed to work with this benchmark. The primary architecture of PHYSGYM is depicted in Figure 2. At its heart, PHYSGYM integrates the environments derived from our dataset, including all associated metadata described in Section 3.1. The core components of the PHYSGYM simulation environment are introduced as follows.

**Environments.** These are directly constructed from the 97 curated physics problems, each encapsulating the environment content, equation, and variable definitions. The associated code is transformed into an executable function which accepts numerical inputs for the defined input variables and executes the code to simulate the environment's response, returning the corresponding outputs.

**Interface.** This serves as the control panel for experiments and provides several key functionalities:
- *Quota Management*: Allows for setting limitations on experimental resources, such as the number of experiments or simulation steps.

- *Environmental Prior Knowledge Control*: Enables varying the level of prior information about the environment that is exposed to the model under test.
- *Historical Data Management*: Tracks and stores the history of interactions, observations, and model hypotheses.
- *Automatic Evaluation Component*: Integrates the evaluation metrics to provide automated assessment of model performance.

### 3.3 Sophisticated Prior Control

A key feature of PHYSGYM is its capacity for sophisticated control over the environmental prior knowledge available to an agent. By meticulously managing the availability of the environment's metadata, we can precisely adjust the level of prior information an agent possesses when tackling a problem. This fine-grained control allows for the systematic observation of differences in agent behavior and performance under varying degrees of physical priors. Importantly, this enables us to investigate how agents balance deductive reasoning (leveraging prior knowledge) with inductive learning (from new interactions), and how they navigate the interplay between theoretical understanding and experimental exploration. For instance, control can be exerted over:

- The visibility of the textual problem description (`context`).
- The availability of physical descriptions for `input_variables` and the `output_variable`.
- Whether variable names adhere to common-sense physical conventions (e.g., $m$ for mass, $v$ for velocity) or are obfuscated (e.g., $var_1$, $var_2$).

Figure 1 illustrates examples of four levels of priors that can be configured within PHYSGYM:

- **Level 1**: In this setting, the task is simplified to a reasoning problem where full observational access and testing capabilities are permitted. The agent is provided with rich contextual information, and solutions can, in theory, be derived entirely through deduction based on the provided priors.
- **Level 2 & 3**: These levels represent a spectrum where partial information is provided. For example, in Level 2, variable descriptions are available but not the full problem context, challenging the agent to integrate incomplete priors with active exploration.
- **Level 4**: At this end of the spectrum, PHYSGYM-L4 degenerates into an interactive equation discovery environment. Most explicit prior knowledge (like textual descriptions or meaningful variable names) is withheld. The primary implicit prior is the understanding that the underlying relationships are governed by equations pertinent to the physical world.

It is important to note that these four levels are illustrative examples and do not encompass all possible configurations. By selectively masking or revealing different combinations of metadata fields and variable information, a more extensive array of nuanced prior knowledge settings can be created to probe specific aspects of agent learning and reasoning. This flexibility is paramount for studying how agents adapt to and leverage different informational landscapes.

### 3.4 Evaluation Metrics

The primary evaluation metric for model performance is the **Success Rate**. This binary metric awards one point for each correctly solved task, with the final score representing the percentage of successfully completed tasks across the entire test set. Success Rate serves as the foundation for both comparative analysis between models and for measuring performance degradation when varying the prior information provided to the models.

Each task is considered solved if the equation proposed by the model is equivalent to the ground-truth equation of the task. We employ two evaluation methods to determine this equivalence. The primary method, SymPy-based Symbolic Evaluation, rigorously verifies the mathematical equivalence between candidate and ground-truth equations. However, this method occasionally produces false negatives when encountering LLM-generated notations (e.g., np.pi) that SymPy fails to properly parse. To mitigate this limitation, we supplement with LLM-based Equivalence Assessment, which leverages an LLM to judge equation equivalence. A task is considered successfully solved if either evaluation method confirms equivalence.

Additionally, PHYSGYM incorporates a suite of auxillary evaluation metrics to assess the performance of models in details.

**Consistency Metrics.**  A fundamental sanity check for our evaluation framework is whether the model's proposed equation remains consistent with previously observed data points. Since the model generates hypotheses based on its proposed variable valuations and corresponding outputs, we expect mathematically sound models to formulate equations that accurately fit these observations. To quantify this consistency, we employ multiple statistical metrics that measure the alignment between the proposed equations and the observed data points. These include the Coefficient of Determination ($R^2$), Mean Squared Error (MSE), Kendall's Tau ($\tau$) rank correlation, and Mean Absolute Percentage Error (MAPE).

**Task Difficulty Metrics.**  To facilitate a detailed analysis of the performance of the model, we incorporate quantitative metrics of the difficulty of the task. We employ two complementary heuristics to assess the complexity of ground-truth equations: equation length (measured by character count) and variable count. Our approach rests on two key observations about mathematical complexity: First, when comparing equations of equal length, those containing more variables typically present greater estimation challenges due to increased dimensionality of the problem space. Second, when the number of variables is constant, longer equations generally indicate more complex mathematical relationships, involving additional operations or nested functions that are inherently more difficult to model.

## 4   Experiments

In this section, we evaluate representative LLMs in the PHYSGYM benchmark and analyze the results.

### 4.1   Experimental Setup

We implement a basic prompt-based method in which the prompt describes the objective of the task and specifies the required input and output data structures. The LLM is instructed to generate three key components: (1) a specification of proposed experiments consisting of a list of parameter valuations; (2) a current hypothesis formulated by the model; and (3) a Boolean flag that determines whether the current hypothesis warrants testing, which shows the model's confidence in how well its hypothesis captures the underlying mechanism. The complete prompt is provided in the Appendix E.

Our baseline agent implements a fully interactive framework that maintains a complete history of experimental observations, hypotheses, and test results. While the agent generates its plan from a single prompt per turn, this design allows for flexible complexity: the agent can propose single experiments or sophisticated multi-experiment plans. This approach addresses the challenge of managing LLM context windows, which is a non-trivial problem in multi-turn agent design.

**Oracle Test and Experiment Quota.**  Throughout the experimental phase, the model is permitted to conduct one oracle test and receive its results. These results include a fitness metric and a judgment of symbolic equivalence. The interface will invoke the metrics module to return the test results, which are then incorporated into the experimental records and fed back to the model for the subsequent turn. The total quota for experiments is set to 100.

**Input Information.**  In each turn, the LLM is provided with the following information: (1) the physics context, (2) the symbolic names and natural language descriptions of controllable variables and the observable variable, (3) a history of prior experimental records, and (4) the remaining quotas for experiments and oracle hypothesis tests. The content for (1) and (2) varies across different prior levels. Observational history is included in the experimental records.

**Prior Levels.**  We implement the 4 different prior levels as introduced in Section 3.3, denoted as L1 to L4. In L1 we provide the original `context` and full description of the variables. In L2 substitute `context` with "Unknown context.". In L3 we further use the meaningless description for the variables. In L4 we change the namespace of the variables to $var_1$, $var_2$, etc.

**Models & Configurations.**  We evaluated four distinct commercial models from three different companies: Gemini-2.5-Pro and Gemini-2.5-flash:thinking [57], Claude 3.7 Sonnet [58], and OpenAI o4 mini [59]. Among these, Claude 3.7 Sonnet was configured as a non-reasoning model. We

also included two open-sourced lightweight models: gpt-oss-20b [60] and Qwen3-32B [61]. The temperature coefficient for all LLMs was set to 0.3. The maximum token limits were configured as follows: 50,000 for Gemini models and OpenAI o4 mini, 5,000 for Claude 3.7 Sonnet, and 4,000 for Qwen3 and gpt-oss. The "thinking" level for OpenAI o4 mini was set to "high".

## 4.2 Results and Observations

**Impact of Prior Knowledge on Model Performance.** Intuitively, reducing prior knowledge should increase task difficulty and lower success rates for models that can effectively leverage such information. Our systematic evaluation across four prior knowledge levels confirms this expectation while revealing distinct patterns across LLMs.

As shown in Figure 3, as prior knowledge diminishes, the performance decline was substantial across all models. For instance, o4-mini's accuracy dropped from 62.89% to 41.24% with the removal of contextual information, and performance at the lowest prior level (L4) was roughly half of that at the highest (L1) for all models (Table 1). The non-thinking model, Claude 3.7 Sonnet, showed a particularly interesting profile. It suffered the most pronounced accuracy drop from L3 to L4, indicating a high sensitivity to variable nomenclature. However, it gained minimal benefit from the richer information from L3 to L1, suggesting its ability to leverage complex prior knowledge for reasoning is limited.

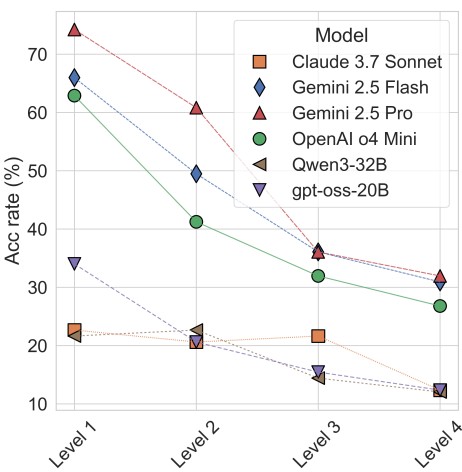

Figure 3: Success rates of different models by prior level.

For individual problems, success patterns are not monotonic. A more detailed analysis (Appendix C.4) of the results from Gemini-2.5-flash, o4-mini, and Claude reveals a non-monotonic inclusion relationship between the sets of problems solved at different prior knowledge levels. Specifically, some problems that are successfully solved at lower prior levels (e.g., L3 or L4) fail at higher prior levels (e.g., L1 or L2), and vice versa. This counterintuitive pattern indicates a fundamental limitation in current models' ability to consistently and rationally utilize prior knowledge across varying information contexts. These findings underscore the complex and non-linear effects of prior knowledge on model reasoning capabilities and validate that our experimental setup provides valuable insights into how models process and integrate information when understanding physical environments.

**Prior is More Important for Difficult Task.** To investigate the relationship between prior knowledge and model performance, we conducted a more granular analysis across varying task difficulty levels. We hypothesized that task difficulty would significantly influence how models leverage prior knowledge when formulating experiments and environmental hypotheses.

For quantitative assessment, we use the number of controllable variables as a heuristic measure of task difficulty. We adopted it as a reasonable proxy following the established principle that higher-dimensional problems present greater modeling challenges. We categorized task dimensionality into four distinct groups: problems with 1-3 variables, 4-6 variables, 7-9 variables, and those with 10 or more variables. Figure 4 illustrates the success rate for 3 models as a function of prior knowledge and difficulty level.

Our analysis revealed several noteworthy patterns. First, performance consistently decreased as difficulty increased, validating our difficulty heuristic. Furthermore, we observed that tasks of varying difficulty responded differently to prior knowledge levels. This distinction was particularly evident in the performance transition from L3 to L4. For moderate difficulty tasks (4-6 variables), this transition produced minimal performance degradation. However, for higher difficulty tasks (7-9 variables), we observed a substantial performance drop between these same prior levels. For environments with 10+ variables, o4 and Gemini only solved the task at L1. This suggests that current models rely on prior knowledge to solve more complex environments, lacking the ability to design effective experiments.

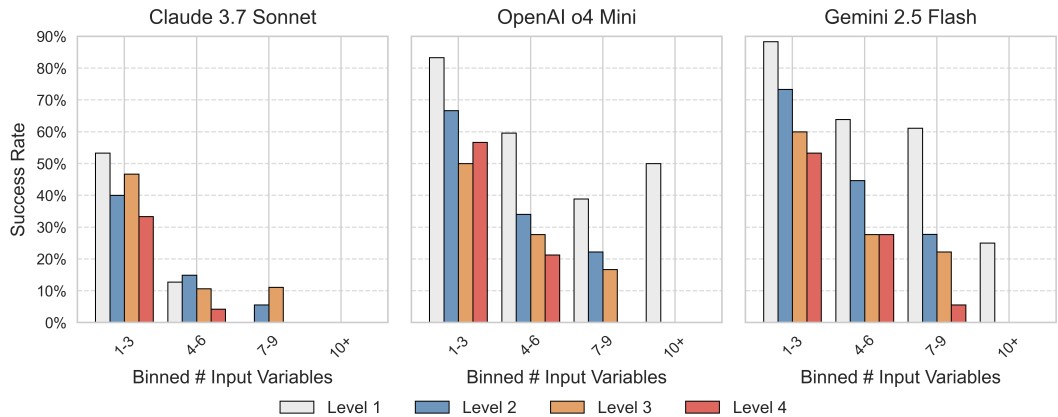

Figure 4: Model success rate as a function of prior knowledge for tasks grouped by dimensionality.

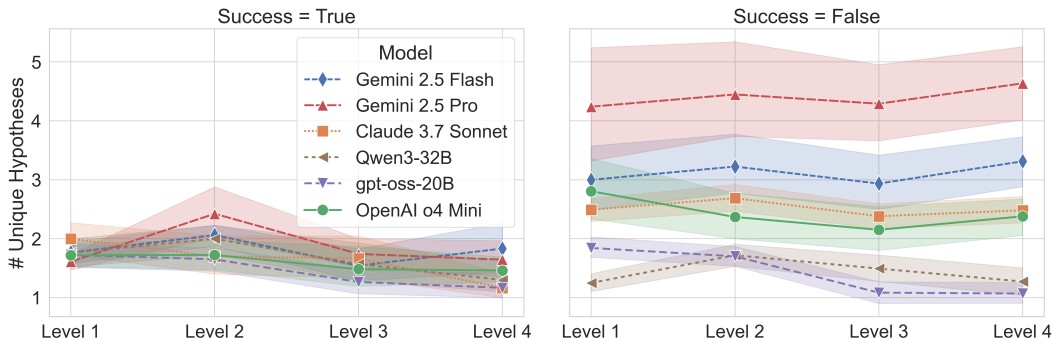

Figure 5: The number of unique hypotheses proposed at different prior levels.

**Unique hypotheses vs prior levels.** The number of distinct hypotheses can reflect the model's ability to dynamically adjust its conjectures in response to experimental outcomes. Figure 5 shows the evolution of the number of unique hypotheses synthesized by the model throughout the entire process. We capture all hypotheses from the model's output, including those flagged as not to be tested. Overall, Gemini models demonstrate the highest ability for hypothesis adjustment. In particular, under conditions of greater uncertainty, they exhibit significant ability to adapt their strategies.

In successful instances, the diversity of hypotheses proposed by Claude and o4 mini diminishes as the prior level increases. This observation further substantiates that, when prior knowledge is limited, Claude's success is not substantially influenced by posterior observations, relying instead heavily on its inherent biases. In contrast, Gemini maintains a comparatively higher diversity of hypotheses at level 4, suggesting that its ability to revise conjectures based on observations possesses stronger generalization capabilities across varying prior levels.

**Exploration Dynamics in Data Collection Under Varying Priors.** As prior knowledge is reduced, all models exhibit a clear increase in the number of interaction experiments in the successful examples, indicating a greater reliance on active experimentation. For example, Gemini-2.5-flash increases its experiment count from 10.6 at L1 to 20.6 at L4, while OpenAI-o4-mini increases from 7.2 to 20.1. This trend reflects a shift from low-effort recognition strategies to more exploratory behaviors as the prior structure is removed. In particular, these increases are accompanied by an increase in turns, tests, and generated hypotheses, suggesting that models not only interact more frequently but also perform more complex reasoning cycles under uncertainty.

Claude-3.7-Sonnet shows a more erratic pattern. Although it increases its samples from L1 to L2, it does not scale its interaction efforts as sharply at L3 and L4. In fact, its sample count slightly declines at L4, and its accuracy remains considerably lower across all levels (e.g., 22.7% at L1 and only 12.4% at L4). This may indicate that Claude is less capable of using feedback loops or fails to appropriately

escalate its exploration strategy when prior structure is unavailable. Full results are shown in Table 1 in the Appendix.

Overall, these results highlight the importance of interactive evaluation settings that vary prior information. They reveal not just whether models can succeed in a task, but how they approach scientific reasoning. In this context, the number of interaction samples becomes a proxy for epistemic engagement: models that scale their behavior with uncertainty are arguably demonstrating a more principled and scientific approach to discovery. This underscores the central motivation behind PHYSGYM: to move beyond static benchmarks and provide a controlled, interactive environment where both reasoning strategies and discovery dynamics can be rigorously assessed.

**Case Studies: When Prior Knowledge Becomes a Constraint.** To illustrate the nuanced effects of prior knowledge on model reasoning, we examine three representative cases that exhibit non-monotonic success patterns. In Environment 310 (relativistic mirror), all models succeeded only at Level 1, failing at lower prior levels. Analysis reveals that without contextual cues about relativistic effects, models adopted overly conservative experimental designs with insufficient $E/m$ ratios, and critically, failed to reason about causal relationships between energy, mass, and velocity changes. Conversely, Environment 716 (rotational speed measurement) was solved by Gemini and o4-mini only at Levels 3 and 4, failing when full context was provided. Here, prior knowledge constrained exploration: models designed experiments using realistic physical values (e.g., Coulomb's constant $\approx 9 \times 10^9$) which, while physically plausible, provided no advantage for formula inference. Environment 409 (electromagnetic tubular field) demonstrates the influence of variable naming: models succeeded at L1-L2 but not L3, with hypotheses at L3 predominantly conforming to the form $\epsilon_0 E_0 f(a/r)$, which is a bias introduced by physical conventions; while at L4 with anonymized variables, models explored different functional forms and eventually converged to correct solutions. These cases underscore a fundamental tension: models struggle to disentangle pattern-matching from mechanistic reasoning, becoming either paralyzed without sufficient priors or constrained by them when available. Detailed case study analyses are provided in Appendix D.

# 5   Conclusion

We introduced PHYSGYM, a benchmark designed to dissect scientific reasoning by systematically controlling prior knowledge in interactive physics environments. Its design enables fine-grained analysis of how models balance deduction with induction—a capability absent in exisiting benchmarks. Our central finding is that prior knowledge is a double-edged sword: models are sometimes constrained by rich context, solving certain problems only when priors are stripped away. This reveals a fundamental conflict between memorized pattern-matching and true mechanistic reasoning. PHYSGYM's interactive setting highlights critical flaws in current exploration dynamics, as models fail to design informative experiments or reason about causality, becoming paralyzed without context or trapped by it. These findings underscore the importance of PHYSGYM for understanding and advancing AI in science, suggesting that fostering robust reasoning and adaptive exploration is key to developing LLMs as effective scientific partners.

Our work opens several promising avenues for future research. First, exploring counterfactual physics would create scenarios where memorized training data becomes a disadvantage, forcing models to rely on interactive reasoning and directly testing the conflict between knowledge recall and discovery. Second, investigating fine-tuning for low-prior discovery tasks could reveal whether interactive scientific reasoning skills can be learned or whether current benchmarks expose fundamental LLM limitations. Third, our findings point to concrete ways to build stronger agents: incorporating exploration modules that maximize information gain and falsify hypotheses; leveraging PHYSGYM's structured levels (L1-L4) for automated curriculum learning; and using our multi-level setup to identify crucial prior knowledge for developing agents that efficiently query knowledge sources.

# Acknowledgment

The research reported in this publication was supported by funding from King Abdullah University of Science and Technology (KAUST) - Center of Excellence for Generative AI, under award number 5940.

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

Table 1: Results overview of LLMs by Prior Level. Acc denotes the success rate. The values in the table show the average number of experiments, tests, runs, unique hypotheses proposed ((U) Hyps), and total hypotheses proposed in successful examples.

| Model | Mode | Acc (%) | Experiments | Tests | Turns | (U)Hyps | Total Hyps |
|---|---|---|---|---|---|---|---|
| Gemini-2.5-flash | L1 | 65.98 | 10.62 | 1.08 | 2.28 | 1.77 | 2.28 |
| | L2 | 49.48 | 12.92 | 1.25 | 2.60 | 2.06 | 2.54 |
| | L3 | 36.08 | 15.51 | 1.23 | 2.77 | 1.54 | 1.86 |
| | L4 | 30.93 | 20.63 | 1.20 | 3.43 | 1.83 | 2.37 |
| OpenAI-o4-mini | L1 | 62.89 | 7.21 | 1.21 | 2.28 | 1.72 | 2.21 |
| | L2 | 41.24 | 12.58 | 1.15 | 3.02 | 1.72 | 2.12 |
| | L3 | 31.96 | 13.90 | 1.19 | 3.10 | 1.48 | 1.68 |
| | L4 | 27.84 | 20.11 | 1.26 | 3.96 | 1.52 | 1.70 |
| Claude-3.7-Sonnet | L1 | 22.68 | 7.50 | 1.27 | 2.36 | 2.00 | 2.36 |
| | L2 | 20.62 | 12.60 | 1.30 | 3.20 | 1.70 | 2.55 |
| | L3 | 21.65 | 12.00 | 1.29 | 2.86 | 1.67 | 1.95 |
| | L4 | 12.37 | 10.92 | 1.00 | 2.33 | 1.17 | 1.17 |
| Gemini-2.5-pro | L1 | 74.23 | 6.49 | 1.10 | 2.14 | 1.61 | 2.14 |
| | L2 | 60.82 | 11.42 | 1.20 | 3.12 | 2.42 | 3.00 |
| | L3 | 36.08 | 12.31 | 1.20 | 2.69 | 1.74 | 1.94 |
| | L4 | 31.96 | 14.10 | 1.13 | 3.03 | 1.65 | 1.84 |
| gpt-oss-20b | L1 | 34.02 | 6.18 | 1.09 | 3.39 | 1.73 | 2.00 |
| | L2 | 20.62 | 6.85 | 1.00 | 3.60 | 1.65 | 1.80 |
| | L3 | 15.46 | 7.53 | 1.13 | 2.60 | 1.27 | 1.40 |
| | L4 | 12.37 | 9.17 | 1.17 | 3.50 | 1.17 | 1.25 |
| Qwen3-32B | L1 | 21.65 | 5.24 | 1.10 | 4.14 | 1.76 | 2.10 |
| | L2 | 22.68 | 8.82 | 1.14 | 4.36 | 2.00 | 2.14 |
| | L3 | 14.43 | 12.50 | 1.07 | 3.64 | 1.57 | 1.71 |
| | L4 | 12.05 | 11.10 | 1.20 | 4.00 | 1.30 | 1.40 |

## A Broader Impacts and Limitations

**Broader Impacts.** As a simulated benchmark, PHYSGYM has no direct societal impact but provides a safe, controlled environment to accelerate AI development for scientific discovery. Our findings reveal that models can inherit biases from training data and that prior knowledge affects them in complex, counterintuitive ways. PHYSGYM provides a systematic framework to measure and understand these effects, contributing to the responsible development of AI systems. As AI becomes more involved in scientific research, understanding these dynamics is crucial for ensuring reliability and trustworthiness.

**Limitations.** A current limitation is the static nature of the problem set within the PHYSGYM benchmark. The existing dataset is fixed and based on manual construction from PHYBench. A key area for future work is the development of methods for automated generation of new physics environments and problem instances to prevent overfitting and enable broader applicability. Moreover, our approach to quantifying task difficulty relies on relatively simple heuristics, such as equation length and the number of variables. While these serve as useful proxies, they are approximations of the true underlying complexity. More sophisticated metrics from the symbolic regression literature could provide more precise characterizations of task difficulty. Additionally, our two-stage evaluation protocol (SymPy followed by LLM-based fallback) introduces some subjectivity, though we mitigate this through structured judgments with confidence scores and detailed explanations.

## B Dataset Statistics

This table presents the statistical overview of our physics dataset, which contains 97 samples distributed across six different fundamental physics domains. Each sample is characterized by an

average of 4.7 input variables and 1.0 dummy variables. The context length varies considerably across domains, with Optics having the longest average context (1241.4 characters) and Mechanics having the shortest (845.9 characters).

Table 2: Dataset Statistics

| Domain | # Samples | # Input Variables | # Dummy Variables | Length of Context (char) |
|---|---|---|---|---|
| Optics | 7 | 4.9 | 0.6 | 1241.4 |
| Mechanics | 39 | 4.1 | 1.2 | 845.9 |
| Electricity | 31 | 5.5 | 1.0 | 1026.8 |
| Thermodynamics | 10 | 4.8 | 1.1 | 885.1 |
| Modern | 7 | 4.3 | 0.4 | 857.4 |
| Advanced | 3 | 5.0 | 0.3 | 1217.7 |
| All | 97 | 4.7 | 1.0 | 948.6 |

## C  Additional Results

### C.1  Full Statistics

Full statistics of the evaluated Large Language Models are provided in Table 1. The table details the performance of each model across the four prior knowledge levels, reporting success rates alongside resource consumption metrics. Specifically, the values show the average number of experiments, tests, runs, unique hypotheses proposed, and total hypotheses generated, all computed over successful task instances.

### C.2  Efficiency Metrics

Beyond accuracy and task complexity, we evaluate how economically the model arrives at its hypotheses. We therefore introduce three complementary efficiency metrics. *Iteration efficiency* quantifies the average number of experimental rounds (i.e., proposals of variable valuations and corresponding evaluations) required for the model to reach a predefined performance threshold on consistency metrics. *Sample efficiency* measures the total count of experiment samples the model consumes before achieving the success, highlighting its ability to learn from limited observations. *Hypothesis efficiency* captures the number of candidate equations the model generates and discards prior to converging on a final hypothesis; lower values indicate more targeted exploration of the hypothesis space. Together, these metrics provide a holistic view of the model's resource requirements in terms of iterations, data samples, and hypothesis evaluations.

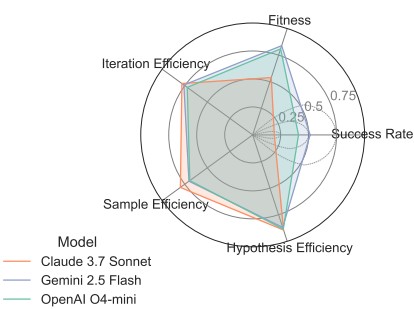

Figure 6: Performance metrics for different models.

Figure 6 presents several performance metrics averaged over all tasks, regardless of level, for various models. The results show that Gemini Flash demonstrates strong performance across all dimensions, while Claude exhibits particular strength in the sample efficiency dimension.

### C.3  Fitness

A fundamental requirement for any scientific reasoning model is that its predictions align closely with past observations. In the ideal case, every data point lies exactly on the hypothesis curve. More realistically, we demand that the model incur a low empirical risk—for example, a small mean squared error or a high coefficient of determination $R^2$ on these observations.

We assess the goodness-of-fit of the model proposed hypothesis functions by measuring the $R^2$ metric on the experiments observed from previous executions. A higher value of $R^2$ indicates a better fit to

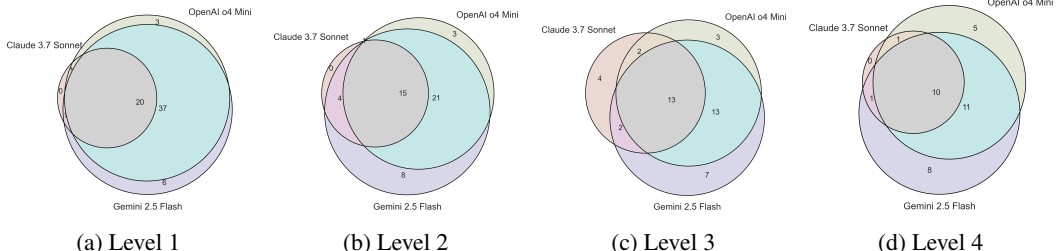

(a) Level 1      (b) Level 2      (c) Level 3      (d) Level 4

Figure 8: Venn diagrams illustrating the overlap of problems solved by 3 models at each level.

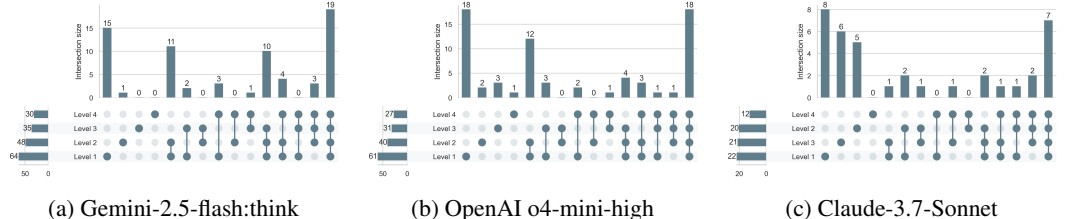

(a) Gemini-2.5-flash:think      (b) OpenAI o4-mini-high      (c) Claude-3.7-Sonnet

Figure 9: UpSet plots illustrating the intersection of problems solved by the model at four levels.

the data, reflecting stronger hypothesis-data consistency. We report the maximum Fit Quality across all hypotheses for each model's complete experimental history at the point when each hypothesis was tested. The results are presented in Figure 7.

Several key observations emerge from this analysis:

**Superior performance of thinking models.** Consistent with the overall success rates, thinking models demonstrate significantly superior Fit Quality compared to the non-thinking model across all prior knowledge levels. This advantage is expected, as thinking models can verify their hypotheses during the reasoning process and adapt them to better align with experimental constraints.

**Prior Knowledge vs. Observations Balance.** Thinking models effectively balance prior knowledge and observations. As prior knowledge is reduced, their fit quality shows only a modest decrease, indicating they can incorporate experimental evidence to form accurate hypotheses. In contrast, the non-thinking Claude model displays a counterintuitive trend: its fit quality worsens with more prior knowledge. This suggests the model anchors too heavily on prior information, failing to integrate conflicting experimental data.

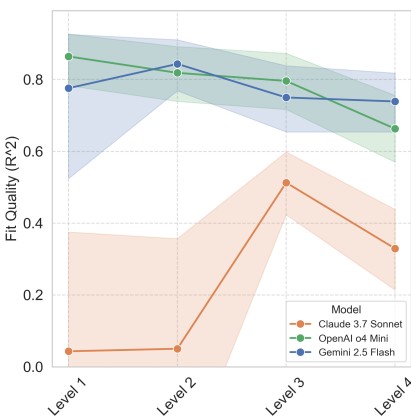

Figure 7: The quality of fit of models' hypotheses and historical experimental results, grouped by prior level.

### C.4 Overlaps between the solved sets

In this section, we aim to explore the following questions: How significantly do different models differ in the sets of problems they can solve? How much overlap exists in the problem sets that the same model can solve under different prior knowledge levels? If a problem is solved at a level with less information, is it more likely to be solved at levels with more information?

Figure 8 shows the relationships between problem sets solved by three different LLMs at each given level. We use Venn diagrams to illustrate the intersections between these sets. Overall, the problem sets solved by Gemini-flash and o4-mini become more divergent as the level increases. At levels 1, 2, and 4, Claude's solved problem set is contained within those of the other two models. Level 3 is an

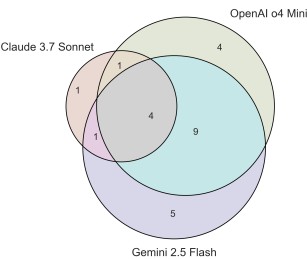

Figure 10: Venn diagram showing the overlap of problems solved by the three models across all levels.

exception. Combined with Figure 7, we observe that Claude's behavior at level 3 differs significantly from the other two models, which may indicate that Claude has stronger symbolic priors.

Figure 9 demonstrates the relationships between success sets of the same model across different levels. We employ UpSet plots, where solid dots below the graph represent included sets and hollow dots represent excluded sets. Several interesting observations emerge: 4 problems were solved only at levels 1, 2, and 4; 3 problems were solved only at levels 1 and 4; and another 3 problems were solved only at levels 2, 3, and 4. The existence of these cases indicates that our assumption does not hold universally—prior knowledge does not always facilitate problem solving. We will conduct detailed case analyses in Section D to examine possible reasons.

Comparing across models, we find that Gemini-flash and o4-mini exhibit similar distribution patterns, while Claude shows significant differences, with many problems solved only at specific levels. This may suggest that Claude's ability to reason based on existing theoretical knowledge is limited, which aligns with expectations given that it is not a reasoning model. Figure 10 further illustrates the differences between models. Even for the set of problems that can be solved across all prior knowledge levels, there are significant differences among the different models.

## D   Case study

In this chapter, we discuss results from specific case studies. These examples demonstrate behavioral differences across various prior knowledge levels and illustrate how prior knowledge influences hypothesis formation and experimental design. They also reveal different capability deficiencies exhibited by models under current baseline implementations. In the first example, Env. 310, we observe that models with incomplete prior knowledge lack the ability to reason about causal relationships between variables and to explore the sample space effectively. In Env. 409, we find that variable namespaces significantly influence the model's hypothesis formation. In Env. 716, prior knowledge actually impairs model performance, as experimental design is heavily influenced by preconceptions about physical quantities. These findings indicate substantial room for improvement in current methods, particularly in effectively leveraging prior knowledge and balancing reasoning with exploration.

For convenience, in the following discussion, we will use these abbreviated names: Gemini-2.5-flash:think will be referred to as Gemini-flash, Gemini-2.5-Pro as Gemini-pro, OpenAI-o4-mini-high as o4-mini, and Claude-3.7-Sonnet as Claude.

### D.1   Environment 310: only solved at Level 1 for all models

**Data**

`context`: Consider an ideal mirror moving at relativistic velocity, with mass $m$ and area $S_0$. (The direction of photon incidence is the same as the direction of the mirror's motion.) Now consider the case where the mirror is moving with an initial velocity $\beta_0 c$. In this situation, the mirror is unconstrained by external forces, and photons are incident on it with constant power for a certain period of time, with energy $E$. Assuming the mirror's velocity after irradiation is $\beta_1 c$, find the expression for $\beta_1$.

`equation` (in LaTeX format):

$$\beta_1 = \frac{\left(\sqrt{\frac{1+\beta_0}{1-\beta_0}} + \frac{2E}{m*299792458^2}\right)^2 - 1}{\left(\sqrt{\frac{1+\beta_0}{1-\beta_0}} + \frac{2E}{m*299792458^2}\right)^2 + 1}$$

`input_variables`:

- $\beta_0$: Initial velocity of the mirror as a fraction of the speed of light (dimensionless, between -1 and 1).
- $E$: Energy of the incident photons (in joules, $J$).
- $m$: Mass of the mirror (in kilograms, $kg$).

`output_variable`:

- $\beta_1$: Final velocity of the mirror as a fraction of the speed of light (dimensionless, between -1 and 1).

`dummy_variables`:

- $S_0$: Area of the mirror (in square meters, $m^2$).

**Discussion**

As provided by PHYBench, the answer can be derived from the following two equations:

$$E + \frac{mc^2}{\sqrt{1 - \beta_0{}^2}} = E' + \frac{mc^2}{\sqrt{1 - \beta_1{}^2}}, \frac{E}{c} + \frac{mc\beta_0}{\sqrt{1 - \beta_0{}^2}} = \frac{mc\beta_1}{\sqrt{1 - \beta_1{}^2}} - \frac{E'}{c}$$

which is straightforward with the knowledge of relativistic effects.

The Level 2 configuration of this problem presents a significant test of the model's ability to utilize existing knowledge for reasoning and experimental design. The key issue lies in the relativistic correction term present in the solution.

In Gemini's initial experimental setup, all observations appeared to conform to the pattern $\beta_1 = \beta_0$. This occurred because the ratio of $E$ to $m$ was too small, resulting in output values virtually identical to $\beta_0$. Throughout subsequent reasoning, the model struggled to overcome the influence of the $\beta_1 = \beta_0$ hypothesis, which repeatedly emerged during Gemini's thinking process. This reveals several critical issues:

First, regarding experimental design, the model did not explore sufficiently extreme scenarios, instead adopting relatively conservative parameter ranges.

Second, **the model failed to investigate potential relationships between controllable variables or attempt to discover underlying causality through experimentation**. The model should have been able to infer possible relationships between incident photons and mirror velocity by examining logical associations in the variable hypotheses, including how energy and mirror mass affect the final outcome. By connecting this to relevant relativistic knowledge, the model should have recognized the probable inclusion of a light-speed term, prompting the use of significantly larger $E/m$ values in experimental settings. Concerning the inaccuracy of the $\beta_1 = \beta_0$ hypothesis, the model merely assumed the existence of a correction term without reasoning through its underlying cause. This indicates substantial room for improvement in the model's ability to reason about potential relationships between known entities in unfamiliar environments.

Furthermore, this example illustrates the fundamental distinction between generating data points through interaction and inductively reasoning from existing data points. The sampling scope employed during data construction naturally incorporates prior knowledge, which is therefore inconsistent with the assumption-free problem setting we establish in Level 4.

## D.2 Environment 409: only solved at Level 1 & 2

In the case of Environment 409, both Gemini-flash and o4-mini successfully derived the correct equation only at levels 1 and 2. Notably, we observed that model performance at level 3 was inferior to that at level 4, suggesting that prior knowledge from symbolic namespaces can sometimes impede the model's posterior observations. We also experimented with Gemini-pro, which similarly failed at level 3 but arrived at the correct conclusion at level 4.

### Data

`context`: In electromagnetism, we often study the problem of electromagnetic field distribution in a region without charge or current distribution. In such cases, the electromagnetic field will become a tubular field. The so-called tubular field is named because of the nature of the velocity field of an incompressible fluid at every instant. Its characteristic can be described using the language of vector analysis as the field's divergence being equal to zero. Another perspective is that if we take the field lines of a vector field $\boldsymbol{F}$ (the tangent line at each point being the direction of field strength) and select a flux tube along a set of field lines passing through a certain cross-section, then at any cross-section of the flux tube, the field flux:

$$\int F \cdot \mathrm{d}S = \Phi$$

will be a conserved quantity. This problem will investigate a special case of a tubular field that is rotationally symmetric around the $z$ axis: using oblate spheroidal coordinates to construct a unique tubular field as an electric field or magnetic field. The so-called oblate spheroidal coordinate system is similar to spherical coordinates, and it is generated by the following coordinate transformations:

$$x = a\mathrm{ch}\mu \cos \nu \cos \varphi$$
$$y = a\mathrm{ch}\mu \cos \nu \sin \varphi$$
$$z = a\mathrm{sh}\mu \sin \nu$$

We want the shape of the field lines of an electric field or magnetic field to precisely follow its generatrix direction, i.e., the tangent direction of the curve where field strength changes with $\mu$ while $\nu, \varphi$ remain fixed (upwards when $z > 0$). Find the charge distribution in the $xy$ plane that can produce this electric field distribution. Assume the field strength near the origin is $E_0$. The dielectric constant is $\varepsilon_0$.

`equation` (in LaTeX format):
$$\sigma = \frac{2\varepsilon_0 E_0 a}{\sqrt{a^2 - r^2}}$$

`input_variables`:

- $\varepsilon_0$: Dielectric constant (permittivity of free space) (in farads per meter, $F/m$).
- $E_0$: Electric field strength near the origin (in volts per meter, $V/m$).
- $a$: Parameter of the oblate spheroidal coordinate system (in meters, $m$).
- $r$: Radial distance from the origin in the xy plane (in meters, $m$).

`output_variable`:

- $\sigma$: Surface charge density in the xy plane (in coulombs per square meter, $C/m^2$).

`dummy_variables`: None

### Discussion

In this case, we examine the detailed responses of Gemini-flash at different levels to analyze the discrepancies in its behavior.

**Level 1**   We present Gemini's level 1 reasoning process in the subsequent colorbox. Through the model's returned reasoning process, we observe extensive iterative repetition in the model's reasoning. Compared to the reference solution, it actually omits several key concepts, such as equipotential surfaces and symmetry. Nevertheless, the final answer is correct, indicating that the interpretability of the reasoning process remains insufficient.

**Level 2**   In the level 2 reasoning process, a critical step occurs when the model hypothesizes a correct form based on physical knowledge: $\sigma = C \cdot \epsilon_0 \cdot E_0 \cdot f(a, r)$. By considering the presence of the disk, it infers the existence of the term $\sqrt{a^2 - r^2}$ in the result, thus obtaining a hypothesis that differs from the correct formula by only a constant in the first test. However, it should be noted that in the experimental design at level 2, the model selects the true physical value for $\epsilon_0$. This behavior actually provides no assistance in formula inference and represents an example of excessive reliance on prior knowledge.

**Level 3**   At level 3, none of the models successfully completed the task, including Gemini-pro. However, the data fitting quality of their proposed models was consistently high.

The answer of the Gemini-flash at the final test, after 20 observations:

$$\frac{8}{\sqrt{3}} \, \epsilon_0 E_0 \sqrt{0.2 \cdot \frac{r}{a} + 0.15}$$

The answer of o4-mini, after 35 observations:

$$\epsilon_0 E_0 \left( 2 + 3.48 \left( \frac{r}{a} \right)^{2.81} \right)$$

They both fit the observed data very well, with MSE less than $1 \times 10^{-17}$.

The answer of Gemini-pro after 33 observations:

$$\epsilon_0 E_0 \left( \frac{\left( 3\sqrt{2} - \frac{4\sqrt{3}}{3} \right) \frac{a}{r} + \left( 4\sqrt{3} - 6\sqrt{2} \right)}{\frac{a}{r} - 1} \right)$$

The MSE is also small, less than $1 \times 10^{-4}$.

Through observation, we find significant variation in the hypotheses proposed by different models, yet interestingly, they predominantly conform to the form $C \cdot \epsilon_0 \cdot E_0 \cdot f(a/r)$. This is likely a bias introduced by the physical variable naming conventions.

**Level 4**   At level 4, the hypotheses proposed by the models differ substantially from those at level 3, no longer containing the $f(a/r)$ term. For example, after 16 observations, Gemini-flash reached the following final hypothesis:

$$\frac{2 \, \text{var}_3}{\sqrt{\text{var}_3^2 - 1}}$$

which achieves an MSE of less than $1 \times 10^{-5}$. If we further allow it continue the experiment, when the number of samples is extended to 26, the model successfully derives the correct function. Notably, Gemini-pro demonstrates superior efficiency, obtaining the correct answer after only 14 observations.

### D.3   Environment 716: only solved at Level 3 & 4

We observed a particularly notable case, Environment 716, where both Gemini-flash and o4-mini successfully solved the problem only at the lower-prior settings of levels 3 and 4. This indicates that the environmental context and semantic descriptions of variables actually impaired model performance.

**Data**

`context`: The principle of a rotational speed measurement and control device is as follows. At point O, there is a positive charge with an electric quantity of Q. A lightweight, smooth-walled insulating

thin tube can rotate around a vertical axis through point O in the horizontal plane. At a distance L from point O inside the tube, there is a photoelectric trigger control switch A. A lightweight insulating spring with a free length of L/4 is fixed at the O end, and the other end of the spring is connected to a small ball with mass m and positive charge q. Initially, the system is in static equilibrium. The thin tube rotates about a fixed axis under the action of an external torque, allowing the small ball to move within the thin tube. When the rotational speed $\omega$ of the thin tube gradually increases, the small ball reaches point A in the thin tube and just achieves radial equilibrium relative to the thin tube, triggering the control switch. The external torque instantaneously becomes zero, thus limiting excessive rotational speed; at the same time, the charge at point O becomes an equal amount of negative charge -Q. By measuring the position B of the radial equilibrium point of the small ball relative to the thin tube thereafter, the rotational speed can be determined. If the distance OB is measured to be $L/2$, determine the rotational speed $\omega$ of the thin tube when the ball is at point B. Express the result using the following physical quantities: Electric charge $Q$, ball's electric charge $q$, mass $m$, length $L$, and Coulomb's constant $k$.

`equation` (in LaTeX format):

$$\omega_B = 4\sqrt{\frac{13kqQ}{23mL^3}}$$

Here's your reformatted content:

`input_variables`:

- $k$: Coulomb's constant (in newton-meters squared per coulomb squared, $N \cdot m^2/C^2$).
- $q$: Electric charge of the small ball (in coulombs, $C$).
- $Q$: Electric charge at point O (in coulombs, $C$).
- $m$: Mass of the small ball (in kilograms, $kg$).
- $L$: Distance from point O to point A in the tube (in meters, $m$).

`output_variable`:

- $\omega_B$: Rotational speed of the thin tube when the ball is at point B (in radians per second, $rad/s$).

`dummy_variables`: None

**Discussion**

The answers obtained by Gemini-pro at levels 1 and 2 are as follows:

$$3\sqrt{\frac{kqQ}{mL^3}}$$

$$3.0072\sqrt{\frac{|kqQ|}{mL^3}}$$

We observe that the experimental data designed for levels 1 and 2 exhibit a particular characteristic: the numerical values chosen for the variables closely approximate their actual values in the physical context. For example, in the following response:

```
{
  "next_experiments": [
    {"k": 8000000000.0, "q": 1e-06, "Q": 1e-06, "m": 0.01, "L": 0.1},
    {"k": 9000000000.0, "q": 0.5e-06, "Q": 0.5e-06, "m": 0.01, "L": 0.1},
    {"k": 9000000000.0, "q": 1e-06, "Q": 1e-06, "m": 0.005, "L": 0.1},
    {"k": 9000000000.0, "q": 1e-06, "Q": 1e-06, "m": 0.01, "L": 0.3},
    {"k": 10000000000.0, "q": 3e-06, "Q": 3e-06, "m": 0.002, "L": 0.05}
  ],
  "test_hypothesis_flag": true,
  "current_hypothesis_formula": "3.0072 * (np.abs(k * q * Q) / (m * L**3))**0.5"
}
```

This choice of numerical values is unnecessary for formula inference. While such settings may be more likely to yield valid numerical results initially, once a hypothesis has been formed, the model should explore a broader range of variable configurations. This demonstrates that prior knowledge can sometimes constrain the model's capacity for exploration and innovation.

**Env 409 Level 1 (1st Response of Gemini-2.5-Flash)**

Thinking...

**Evaluating the Physics**

I'm now considering the user's request, they need me to perform in my areas of expertise: Physics and Experimental Design. This is a good challenge. Let's see how to best approach this.

**Deconstructing the Problem**

I need to understand the user's question about $\sigma$. I have to find a formula relating $\sigma$ to $\epsilon_0$, $E_0$, $a$, and $r$ based on the given electric field. The geometry uses oblate spheroidal coordinates, which adds a layer of complexity, but I think I can handle it. Let's break down the coordinates in the xy-plane.

**Pinpointing the Geometry**

I'm analyzing the user's requirements, specifically the request for the electric field in the xy-plane, examining the oblate spheroidal coordinates. I am zeroing in on the cases where the z-coordinate is zero, and the relationships between $\mu$, $\nu$, and $r$. This will clarify the charge distribution.

**Refining the Analysis**

I'm re-examining the geometry, particularly how the xy-plane is represented with oblate spheroidal coordinates, and how the origin (r=0) is approached. Considering mu and nu, and their ranges for the xy-plane, I will determine what the formula $\sigma$ should depend on, relating to $\epsilon_0$, $E_0$, $a$, and $r$.

**Conceptualizing the Approach**

I must use my knowledge of physics and experimental design, starting with the user's request. I need to derive $\sigma$ based on the input variables ($\epsilon_0$, $E_0$, $a$, $r$) and the context of the xy-plane. I have to keep in mind, that based on the problem statement, the xy-plane represents a boundary with zero divergence.

**Calculating Surface Charge Density**

I must use the user's specifications: my expertise, electric fields, oblate spheroidal coordinates, and z=0 plane charge density. Considering the divergence-free field, I am focusing on how the electric field behaves at the origin (r=0) as I zero in on the geometry. I'm relating the xy-plane to the coordinate ranges of $\mu$.

**Formulating the Equations**

I must understand the user's instructions about '$\sigma$' in the xy-plane. I have to connect $\sigma$ to the given variables, $\epsilon_0$, $E_0$, $a$, and $r$, using the electric field described in oblate spheroidal coordinates. Let's see about the implications for the boundary conditions.

**Re-evaluating Coordinate Ranges**

I'm using the user's constraints, particularly the requirement to determine $\sigma$ in the xy-plane. Understanding the coordinate system is vital. I'm refining the coordinate mappings for the xy-plane, focusing on the radial distance $r$ concerning $\mu$ and $\nu$, and how they relate to the disk and the exterior plane.

**Revising the Approach**

I'm analyzing the user's request, where they need me to calculate $\sigma$. I've to consider the xy-plane as a boundary with zero divergence, alongside the given field strength and the variables. I'm now refining my coordinate analysis, looking for the origin.

**Addressing the Boundary**

I'm now focusing on the user's request, determining the surface charge $\sigma$ in the xy-plane using the given variables. Analyzing the coordinate system reveals two cases: inside the disk, $r \le a$ and outside the disk, $r > a$. The geometry dictates that the coordinate ranges will be crucial for determining the charge density.

**Focusing on Boundary Conditions**

I'm checking the user's request, which requires me to derive $\sigma$. I'm thinking about the problem in the xy-plane, considering the divergence-free nature of the electric field. This informs how the electric field at the origin must align.

**Reframing the Question**

I'm focusing on the user's request: finding the surface charge $\sigma$ in the xy-plane. I'm relating it to the known variables: $\epsilon_0$, $E_0$, $a$, and $r$. The goal is to find a physical model that satisfies all constraints.

**Defining Surface Charge Density**
I must use my physics expertise, including the user's input, such as $\sigma$, to determine the surface charge density. The problem states an electric field with zero divergence, especially within the xy-plane. This, alongside the given variables, sets the stage for calculations.

**Decomposing the Field Equations**
I'm carefully analyzing the user's query: I am requested to find $\sigma$ as it relates to the given $E_0$, $\epsilon_0$, $a$, and $r$. I need to connect it to the z=0 plane by assessing the boundary behavior of electric fields. This problem involves the xy-plane.

**Rethinking the Problem**
I'm analyzing the user's request, their question about finding surface charge density, $\sigma$, and I'm trying to link this to the xy-plane. Focusing on how the $E_0$, $a$, and $r$ are behaving within the constraints of the problem.

**Reviewing Coordinate Systems**
I'm re-evaluating the user's request to find sigma and the given variables: $\epsilon_0$, $E_0$, $a$, and $r$. I need to relate them to the electric field in the xy-plane and consider coordinate system variations. Then, I will check for boundary conditions.

**Calculating the Charge Density**
I must apply my knowledge of physics and experimental design to determine the surface charge density, $\sigma$. I'll focus on the user's input, the variables $\epsilon_0$, $E_0$, $a$, and $r$, keeping in mind the z=0 plane and the divergence-free nature of the electric field.

**Verifying Field Direction**
I'm considering the user's instructions, to construct a tubular field and to find $\sigma$ at z=0. I'm now re-evaluating the coordinate system to confirm the field direction and how it relates to the xy-plane.

**Rethinking Boundary Conditions**
I'm analyzing the user's request, reviewing the xy-plane, and the need to find $\sigma$. I need to apply boundary conditions at z=0, to the given variables and the electric field's behavior. I also must keep the tubular field constraint in mind.

**Focusing on the Solution**
I'm re-examining the user's prompt, needing $\sigma$ in the xy-plane using the given parameters. Analyzing the coordinate system reveals two cases: inside the disk ($r \leq a$) and outside the disk ($r > a$). My next step is to investigate the field direction relative to the z-axis.

**Finalizing Charge Distribution**
I need to connect the user's parameters with $\sigma$. I need to determine the electric field's normal component near the disk. I am now finalizing how the divergence-free constraint affects the boundary conditions at z=0. I will apply Gauss's law near the disk.

**Rethinking the Problem's Core**
I am re-evaluating the user's query and its constraints. I need to determine sigma in the xy-plane using the electric field and the given variables. I must now carefully consider boundary conditions, and the implications of divergence. This may impact how I analyze the geometry.

**Pinpointing Key Parameters**
I'm re-examining the user's query and the data. Considering the boundary conditions, field strength, and the goal of finding sigma in the xy-plane. The origin and how to model the disk is also important.

**Calculating the Charge Distribution**
I need to find $\sigma$ given the physics context and the user's request. With the electric field equations established, I must now focus on applying the correct boundary conditions at the z=0 plane. I am seeking to relate the given values for $\epsilon_0$, $E_0$, $a$, and $r$ to the surface charge density. I'll start by applying Gauss' law.

**Deriving Surface Charge Density**
I'm breaking down the user's request: to find the formula for $\sigma$, relating it to the given variables. I have an electric field with zero divergence, therefore I will consider how the field's direction aligns with the z-axis in both regions, considering their individual coordinate ranges.

**Finalizing the Calculation**
I need to derive $\sigma$ given the data, analyzing the electric field, the xy-plane, and boundary conditions. I am now working through the disk ($r \leq a$) and the plane ($r > a$). I will apply Gauss' Law to find a final formula.

**Revising the Approach**

Based on the user's prompt, I'm re-examining how to calculate $\sigma$ within the xy-plane. I must consider the field strength and the specified variables. Also I need to account for the boundary conditions at z=0, using coordinate systems.

**Solving for Density**

I'm working on the user's question, focused on calculating the surface charge density $\sigma$ in the xy-plane. The electric field and the problem geometry, involving the parameters $\epsilon_0$, $E_0$, $a$, and $r$, and the boundary conditions, are all key to the solution. I'm now analyzing and comparing the electric fields above and below the plane.

**Deriving a Solution**

I must understand the user's prompt to connect $\sigma$ to the knowns: the electric field, and parameters. I will use my physics expertise, especially Gauss's law and the electric field's boundary conditions. I'm starting by finding $\sigma$ in the z=0 plane and its relationship to the disk.

**Deriving a Field Model**

I'm currently studying the user's details: their request to determine the surface charge density, $\sigma$, in the xy-plane using the electric field and $\epsilon_0$, $E_0$, $a$, and $r$. I must assess boundary conditions at z=0, while considering coordinate systems. I will use Gauss' law for this calculation.

**Pinpointing Surface Charge**

I must solve for $\sigma$ within the xy-plane using the known values and my expertise. Considering the tubular electric field, the origin, and the xy-plane boundary, I need a way to apply Gauss' law.

**Formulating a Solution**

I need to relate the user's inputs ($\epsilon_0$, $E_0$, $a$, $r$) to $\sigma$ within the xy-plane's conditions. The boundary behavior of the electric field at z=0 is also relevant, alongside the specified electric field. Let's apply Gauss's law.

**Refining the Calculation**

I need to use my knowledge of physics and experimental design, considering the user's request for $\sigma$ and given variables. Calculating the surface charge density in the xy-plane given the field. This also involves understanding the geometry and boundary conditions in order to formulate the solution.

**Calculating the Charge**

I'm evaluating the user's prompt about $\sigma$, tying it to my expertise and the given variables. The challenge is to derive a correct formula for the xy-plane. Focusing on boundary conditions at z=0 and the electric field's divergence. The origin (r=0) and how it approaches $\mu$ is also important.

**Formulating a Model**

I need to find a formula for $\sigma$ in the xy-plane using $\epsilon_0$, $E_0$, $a$, and $r$. I'm focused on how the xy-plane behaves, and the electric field near the origin, considering both cases, when $r < a$, and when $r > a$. The field lines must satisfy zero divergence, which impacts my understanding of the coordinate system.

**Deriving a Final Solution**

I'm finally focused on answering the user. The question is $\sigma$, and I will use the constraints: the electric field, the given variables, boundary conditions, and especially the geometry in order to determine a final formula. I will focus on the xy-plane, particularly applying Gauss' Law, to determine the charge.

**Finalizing the Calculation**

I need to derive $\sigma$ in the xy-plane given the specified values, analyzing the electric field, and using the boundary conditions. I am now working through the inside and outside cases to apply Gauss' Law and finalize the formula.

**Formulating a Final Solution**

I will use my knowledge of physics and experimental design to address the user's request. I'm to use the electric field and the known variables in order to find $\sigma$ on the xy-plane. The disk and exterior plane have separate formulas for this value.

**Calculating the Surface Charge**

I must analyze the user's input. I am to use the equations to find $\sigma$ in the xy-plane, applying the electric field and the known variables. The approach will involve applying boundary conditions at z=0. I will look for the discontinuity to apply Gauss' law.

**Confirming Charge Distribution**

I need to find the formula for $\sigma$ in the xy-plane based on the user's electric field description and the variables. I am focusing on Gauss' law and am considering how the boundary conditions at z=0 affect the electric field and the surface charge.

**Deriving a Formula**

I must fulfill the user's request for $\sigma$, linked to the provided parameters. I am now deriving the field equations and boundary conditions at $z = 0$, and I'm focusing on applying Gauss' law to the xy-plane. This will lead to a formula for my tests.

**Finalizing the Formula**

I must derive the formula for $\sigma$ in the xy-plane with my physics knowledge, the electric field and boundary conditions. The formula, derived from the known variables, will define the charge.

**Formulating a Hypothesis**

I need to derive a formula for $\sigma$ based on the user's request, using the electric field, boundary conditions, and the given values. After careful analysis, I will test my understanding and refine the model.

**Refining the Calculation**

I need to derive a formula for $\sigma$ based on the user's request, using the given electric field and variables. The xy-plane and its implications for the boundary conditions will guide me in this. Considering the known values, I'm aiming for a precise equation.

**Formulating a Final Solution**

I have to find $\sigma$ from the user's request by assessing the xy-plane and utilizing my physics knowledge. Gauss's law will be essential. I'll derive a formula, ensuring it aligns with the electric field's behavior and boundary conditions, starting with the well-known charged disk formula.

I will now test the values, to verify my hypothesis.

**Deriving a Solution**

I must create a formula for $\sigma$ tied to $\epsilon_0$, $E_0$, $a$, and $r$, while keeping the xy-plane and its special boundary in mind based on the user's request. I will be using the information to produce the solution. I will use the electric field and boundary conditions.

**Formulating a Final Solution**

I have to use the user's instructions to determine $\sigma$ using my physics knowledge. I'll determine the formula, considering the electric field, xy-plane, and all given variables and boundary conditions. Gauss's law will be useful. I have determined the proper charge density formula.

**Formulating the Formula**

I'm analyzing the user's input, considering the desired formula relating $\sigma$ to $\epsilon_0$, $E_0$, $a$, and $r$. The xy-plane, with its specific boundary conditions, is crucial. Now, I am focusing on the final formula, a key step.

**Confirming the Field Model**

I'm now ready to provide the formula for the user. My analysis has led to a formula for the charge density $\sigma$ as a function of the given values. The electric field, with its boundary conditions, and the xy-plane has been key, especially by applying Gauss' law. I'll now test this formula.

**Refining the Equation**

I need to calculate $\sigma$ given the user's specifications, considering that I'm an expert in Physics and Experimental Design, and the $\sigma$ formula must be based on the electric field. I've chosen a formula for the observable values, using $\epsilon_0$, $E_0$, $a$, and $r$.

**Deriving a Formula**

I'm analyzing the user's input, requesting $\sigma$ for the xy-plane based on the given values. I'm relating this to the boundary conditions and the z=0 plane to define the charge density. I will use Gauss's Law in my final step.

**Refining the Calculation**

I need to derive a formula for $\sigma$ based on the user's input, using the given electric field and variables. I'm finding the formula relating $\sigma$ to $\epsilon_0$, $E_0$, $a$, and $r$, with the xy-plane's geometry in mind. I am now preparing to test and evaluate my model, using the given information.

**Formulating a Plan**

I am evaluating the user's prompt about calculating $\sigma$, given the parameters, and am considering the conditions of the xy-plane. I must use my understanding of physics and experimental design. Let's apply Gauss's Law.

**Refining the Derivation**

I'm reviewing the user's request, focused on the calculation of surface charge density in the xy-plane. I'm connecting this to the variables given, and making sure the boundary conditions are considered and accurately represent the physics. Then I will formulate the correct formula.

**Calculating the Formula**

I'm working on the user's specifications, particularly $\sigma$ in the xy-plane. I'll be using the formula, relating to $\epsilon_0$, $E_0$, $a$, and $r$, and the condition $r < a$ for this calculation. After assessing the theory I'm ready to propose the formula.

**Response**:

```
{
  "next_experiments": [
    {"epsilon_0": 1, "E_0": 1, "a": 1, "r": 0},
    {"epsilon_0": 1, "E_0": 1, "a": 1, "r": 0.5},
    {"epsilon_0": 1, "E_0": 1, "a": 1, "r": 0.8},
    {"epsilon_0": 1, "E_0": 1, "a": 1, "r": 0.9},
    {"epsilon_0": 1, "E_0": 1, "a": 2, "r": 1},
    {"epsilon_0": 1, "E_0": 1, "a": 0.5, "r": 0.25},
    {"epsilon_0": 1, "E_0": 2, "a": 1, "r": 0.5},
    {"epsilon_0": 1, "E_0": 0.5, "a": 1, "r": 0.5},
    {"epsilon_0": 2, "E_0": 1, "a": 1, "r": 0.5},
    {"epsilon_0": 0.5, "E_0": 1, "a": 1, "r": 0.5}
  ],
  "test_hypothesis_flag": false,
  "current_hypothesis_formula": "2 * epsilon_0 * E_0
  / (1 - (r/a)**2)**0.5"
}
```

---

### Env 409 Level 2 (1st Response of Gemini-2.5-Flash)

**Deducing Variable Relationships**

I'm analyzing the user's request, specifically the `problem_description` lack of context. The goal is to determine the formula connecting $\sigma$ to $\epsilon_0$, $E_0$, $a$, and $r$. Considering that $\epsilon_0$ is fundamental and $E_0$ is electric field strength. The parameter $a$ seems significant.

**Hypothesizing & Unit Checks**

Analyzing the user's parameters to extract a formula relating $\sigma$ to the other variables. Using unit analysis, I see that multiplying $\epsilon_0$ by $E_0$ results in the correct units for $\sigma$. Next, I'll check how $a$ and $r$ might fit.

**Revisiting Formula Derivation**

Given the user's input on $\sigma$, $\epsilon_0$ and $E_0$, I'll focus on refining the formula. From the units, multiplying $\epsilon_0$ and $E_0$ gives us $\sigma$, so this part is correct. Now, I'll consider how $a$ and $r$ fit the formula for the specific geometry.

**Final Formula Consideration**

The user is asking for the formula relating $\sigma$ to the provided variables. My initial hypothesis is $\sigma = C \cdot \epsilon_0 \cdot E_0 \cdot f(a, r)$. Analyzing a conducting disk in an electric field, I suspect a $\sqrt{a^2 - r^2}$ component. Time to refine the equation.

**Refining & Initial Hypothesis**

Considering the user's parameters. Given the electric field $E_0$, dielectric constant $\epsilon_0$, characteristic length $a$, and radial distance $r$ and knowing that the observable variable $\sigma$ is surface

charge density. Combining $\epsilon_0$ and $E_0$ yields $\sigma$. I hypothesize that $\sigma = C \cdot \epsilon_0 \cdot E_0 \cdot f(a, r)$, considering a conducting disk.

**Re-evaluating Hypothesis Formulation**

Based on the user's inputs, specifically $\epsilon_0$, $E_0$, $a$ and $r$, plus my initial hypothesis. I'll try refining my approach, as the initial hypothesis may be too simplistic. Using the units from earlier, I'll reformulate to create the expected observable, $\sigma$.

**Confirming Initial Equation**

I'm analyzing the relationships between $\sigma$, $\epsilon_0$, $E_0$, $a$, and $r$, using $\epsilon_0 \cdot E_0$. My hypothesis:

$$\sigma = \frac{\epsilon_0 \cdot E_0 \cdot a}{\sqrt{a^2 - r^2}} \tag{1}$$

as a first pass. I'll use this to guide experimental design.

**Validating & Experiment Planning**

Given the user's variables, including $\sigma$, $\epsilon_0$, and $E_0$. Considering the geometry mentioned, I will use the initial formula $\sigma = \epsilon_0 \cdot E_0 \cdot a/\sqrt{a^2 - r^2}$. I'll set the test flag to "false" and design 5 initial experiments.

**Formulating the Equation**

Given the user's variables, particularly $\sigma$, $\epsilon_0$, and $E_0$, and incorporating the geometry of a conducting disk with radius $a$ and distance $r$, my initial hypothesis is:

$$\sigma = \frac{\epsilon_0 \cdot E_0 \cdot a}{\sqrt{a^2 - r^2}} \tag{2}$$

**Response**:

```
{
  "next_experiments": [
    {"epsilon_0": 8.854e-12, "E_0": 500, "a": 0.2, "r": 0.1},
    {"epsilon_0": 8.854e-12, "E_0": 1000, "a": 0.2, "r": 0.1},
    {"epsilon_0": 8.854e-12, "E_0": 500, "a": 0.4, "r": 0.2},
    {"epsilon_0": 8.854e-12, "E_0": 500, "a": 0.2, "r": 0.18},
    {"epsilon_0": 8.854e-12, "E_0": 500, "a": 0.2, "r": 0.02}
  ],
  "test_hypothesis_flag": false,
  "current_hypothesis_formula": "epsilon_0 * E_0 * a
  / np.sqrt(a**2 - r**2)"
}
```

# E  Prompt

**Researcher Prompt**

```
# Core Task
Based on the provided problem description, controllable physical
    quantities, observable physical quantities, and historical
    experimental records, your primary goal is to:
1. Deduce the mathematical expression (formula) that describes how
    the observable quantity changes with the controllable physical
    quantities
2. Propose the next set of experimental parameters to test
3. Determine if it's appropriate to formally test your current
    hypothesis

# Inputs
The input will be a JSON dictionary with the following five parts:
```

1. **`problem_description`**: (String)
   * A brief description of the physical phenomenon or system under study
   * Example: "Investigating the relationship between the extension of a spring and the applied force."

2. **`controllable_variables`**: (Dictionary)
   * Physical quantities that can be actively changed in an experiment
   * The dictionary contains:
     * Keys: The `name` of the physical quantity (e.g., "F", "k")
     * Values: Detailed description of the physical quantity
   * Example:
     ```json
       {"F": "The applied force on the spring in Newtons (N)", "k":
     "The spring constant in Newtons per meter (N/m)"}
     ```

3. **`observable_variable`**: (Dictionary)
   * The physical quantity measured in the experiment
   * This quantity changes in response to changes in controllable variables
   * Example:
     ```json
     {"x": "The extension of the spring in meters (m)"}
     ```

4. **`historical_experiments`**: (List of Dictionaries)
   * Data from previously conducted experiments
   * Each dictionary represents a single experimental record
   * Contains all controllable variables and their set values, plus observed results
   * Example:
     ```json
     [
       {"F": 0.5, "k": 10, "x": 0.05},
       {"F": 1.0, "k": 10, "x": 0.10}
     ]
     ```
   * May be an empty list `[]` for the first run

5. **`quota`**: (Dictionary)
   * Defines the remaining budget for experimentation and hypothesis testing.
   * experiments_quota: (Integer) The maximum total number of remaining experimental runs you can try.
   * test_quota: (Integer) The remaining number of times you can set the `test_hypothesis_flag` to true.
   * Example:
     ```json
     {"experiments_quota": 10, "test_quota": 2}
     ```

# Outputs
You must output a JSON dictionary with the following three parts:

1. **`next_experiments`**: (List of Dictionaries)

* Your designed input variable combinations for the next
  experiment(s)
* Each dictionary represents a set of experimental parameters
* Keys are the `name` of controllable physical quantities
* Values are your proposed settings within allowed ranges. Use
  floats or integers.
* Strategy:
  * For empty historical data: Provide exploratory initial
  experimental points
  * For existing data: Select the most informative points to test
  your hypothesis
  * Ensure experimental diversity and coverage

2. **`test_hypothesis_flag`**: (Boolean)
   * `true`: Your current hypothesis is mature and next experiments
   aim to validate it
   * `false`: You're in exploratory phase with low confidence,
   seeking more data

3. **`current_hypothesis_formula`**: (String)
   * Your conjectured mathematical expression relating observable
   quantity to controllable quantities (in Python syntax with basic
   operators)
   * Use the exact variable names defined in the input
   * IMPORTANT: Use "**" operator for powers/exponents. Only if it is
   necessary, you can use "np.xxx" to implement special functions,
   e.g. "np.cos".
   * Examples: "F / k", "a * (F ** 2) + b * L + c"
   * Return `None` or `""` if data is too scarce to form a meaningful
   hypothesis

# Example (First interaction without historical data)

**Input:**
```json
{
  "problem_description": "Investigating the relationship between the
    extension x of an ideal spring (within its elastic limit) and the
    applied force F. The spring constant is k.",
  "controllable_variables": {"F": "The applied force on the ideal
    spring in Newtons (N).","k": "The spring constant in Newtons per
    meter (N/m)."},
  "observable_variable": {"x": "The extension of the spring in meters
    (m)."},
  "historical_experiments": [],
  "quota": {"experiments_quota": 10, "test_quota": 2}
}
```

**Expected Output:**
```json
{
  "next_experiments": [
    {"F": 0.5, "k": 10},
    {"F": 1.0, "k": 10},
    {"F": 2.0, "k": 10},
    {"F": 1.0, "k": 20},
```

```
      {"F": 1.0, "k": 5}
   ],
   "test_hypothesis_flag": false,
   "current_hypothesis_formula": "F / k"
}
```