# OpenReview forum: "PhysGym: Benchmarking LLMs in Interactive Physics Discovery with Controlled Priors"
_NeurIPS.cc/2025/Datasets_and_Benchmarks_Track — NeurIPS 2025 Datasets and Benchmarks Track poster_

### Official Review · Reviewer_iUAt · 2025-07-02

**Rating:** 5
**Confidence:** 3

**Summary:**

This paper introduces PHYSICIST, a novel interactive benchmark and simulation environment designed to evaluate the scientific reasoning capabilities of Large Language Model (LLM) agents. The benchmark's core contribution is its sophisticated control over the level of prior knowledge provided to the agent, allowing for a systematic analysis of how agents discover physical laws under varying degrees of uncertainty. The dataset consists of 97 physics problems derived from PHYBench, each packaged with a textual context, a ground-truth symbolic equation, and an executable Python simulator. The environment allows agents to actively probe these simulations, gather data, and form hypotheses, which are then evaluated for symbolic equivalence. Baseline results for three large commercial LLMs demonstrate that the benchmark can effectively differentiate model performance as prior knowledge is progressively removed across four distinct levels.

**Additional Feedback:**

PHYSICIST is a benchmark with great potential, and its core idea of controlled priors is an important contribution. However, the submission in its current state has significant compliance and documentation gaps. I would be willing to raise my score if the authors address these issues.

- Clarify and Unify the License: Please audit all public-facing materials (paper, OpenReview, GitHub, metadata files) and ensure they all state the same, correct license. This is a top-priority item to resolve any ambiguity for potential users.
- Add an Agentic Baseline: To validate the benchmark's interactive design, please include at least one baseline that performs sequential, step-by-step reasoning and experimentation (e.g., using a ReAct-style framework). This would better demonstrate the environment's full potential.
- Provide Task Generation Scripts: To address the limited scale and originality, could you provide the scripts and methodology used to convert PHYBench problems into your interactive format? This would allow the community to expand the benchmark and prevent overfitting.
- Quantify Verification Quality: Please add details about the manual verification process, such as the number of reviewers or the error rate found during checks.
- Add a Broader Impact Section: Please add a section to the paper briefly discussing the societal impacts and limitations (e.g., explaining why the answer of "10. Broader impacts" is [NA].).

**Dataset Code Accessibility:**

Partly

**Dataset Code Comments:**

- The Croissant metadata file is missing essential fields (name, license, url, etc.) and has no field-level descriptions.
- The README explains how to use the simulators but lacks clear end-to-end instructions for running the full interactive evaluation loop with an agent.

**Ethical Comments:**

No major ethical concerns that would prevent publication were identified, as the dataset deals with abstract physics problems and contains no personal or sensitive information. However, the paper is missing a "Broader Impact" discussion, which is a required component of the NeurIPS checklist. The authors should briefly discuss potential societal impacts, which could include.

**Ethical Considerations:**

No, there are no or only very minor ethics concerns

**Final Justification:**

I have raised my score, as my initial concerns have been sufficiently addressed during the discussion period.

My primary concern was the gap between the benchmark's interactive goals and the apparent simplicity of the baseline's reasoning process. The authors' rebuttal clarified this point. They provided a concrete example demonstrating the baseline's capacity for complex multi-turn reasoning, and offered a reasonable technical rationale for their design choice related to LLM context management.

The authors have also committed to valuable improvements, including a more detailed discussion of these design considerations in the revised paper and an enhancement of the framework to support future research.

Given that my main concerns have been resolved through the authors' clear explanations and commitment to improving their work, I now support the paper for acceptance.

**Limitations Weaknesses:**

While the conceptual foundation of PHYSICIST is strong, the current submission has several significant weaknesses, primarily related to compliance, documentation, and experimental scope.

- **Inconsistent and Ambiguous Licensing:** A critical issue is the lack of clarity and consistency in the dataset's license. There may be discrepancies between the license declared on the submission platform (e.g., OpenReview) and the one specified in the repository or code. This ambiguity creates legal uncertainty for users and undermines the benchmark's trustworthiness and reusability. A single, clear, and consistently applied license is essential.
- **Limited Originality and Scale of Dataset:** The benchmark is constructed from only 97 problems, all of which are sourced from an existing dataset (PHYBench). While the interactive reframing is novel, the core data itself is not. This small scale raises concerns about the potential for models to overfit to the benchmark and limits its statistical power for differentiating between top-performing agents.
- **Lack of Detail on Human Verification:** The paper states that the dataset construction was "LLM-assisted" and followed by "rigorous manual verification." However, no details about this verification process are provided. For a high-quality benchmark, information such as inter-annotator agreement rates, the number of verifiers, or error statistics from the verification process is essential to establish the dataset's quality.
- **Limited Scope of Baselines:** The benchmark is designed to evaluate interactive discovery, yet the baselines are limited to single-turn, non-interactive prompting. This is a significant weakness, as it leaves the core interactive capabilities of the environment (e.g., sequential experimentation, hypothesis revision) largely unevaluated and undemonstrated.

**Strengths Contributions:**

The PHYSICIST benchmark is a timely and well-designed contribution that addresses a significant gap in the evaluation of AI for science.

- **Novel and Important Problem Formulation:** The key strength of this work lies in shifting the evaluation of scientific discovery from static equation-finding to an interactive process. The benchmark's central feature—the ability to systematically control prior knowledge (as shown in Figure 1)—is a powerful tool for disentangling an agent's ability to memorize from its ability to perform true mechanistic inference and exploration. This allows for a much deeper and more nuanced assessment of an LLM's scientific reasoning skills than existing static datasets.
- **Well-Architected Simulation Environment:** The paper presents a comprehensive and thoughtfully designed simulation suite (Figure 2). It includes an interface for managing experimental budgets, a controller for modulating prior knowledge, and a robust evaluator that combines rigorous symbolic equivalence checking (via SymPy) with a fallback LLM assessment to handle syntactic variations. This architecture provides a solid foundation for future research.
- **Clear Validation of the Benchmark's Utility:** The experiments with baseline LLMs (Section 3.2) effectively demonstrate the value of the benchmark's design. The results show a clear, monotonic decrease in success rate as prior knowledge is stripped away (Figure 3), confirming that the levels of prior knowledge create a meaningful difficulty gradient. The analysis also reveals interesting behavioral differences, such as how models increase their experimentation under uncertainty (Table 1), providing insights that would be invisible in a static benchmark.
- **Good Reproducibility Artifacts:** The authors provide the complete dataset, simulation code, and prompts in the supplementary material. The dataset is provided in a structured JSON format with accompanying Python simulators, which is a practical and usable format.

---

> ### Author Rebuttal · Authors · 2025-07-30
>
> We would like to thank the reviewer for their thorough and constructive feedback, and for recognizing the value and contribution of PHYSICIST. We address each point raised by the reviewer below:
>
> ---
> 1. **Licensing, Metadata, and README**
>
> > Reviewer's Comments: "Inconsistent and Ambiguous Licensing," "Croissant metadata file is missing essential fields," "README... lacks clear end-to-end instructions"
>
> Thank you for these precise and actionable points on compliance and documentation. We agree that these elements are critical for the benchmark's long-term value and usability.
> - Licensing: We appreciate the opportunity to clarify our licensing structure.
>   - Dataset: CC0 1.0 Universal (maximizing accessibility)
>   - Code: MIT License (upon publication)
>   - Paper: CC BY 4.0 (following conference requirements)
> - Croissant Metadata: We will thoroughly verify that our croissant file contains all essential fields. The current file incorrectly lists the dataset license, which we will correct to CC0 1.0.
> - README and Instructions: We agree that the README would benefit from more explicit instructions. We will add a detailed, step-by-step guide for running the full interactive evaluation loop, which is implemented in `baseline_experiment.py` with corresponding scripts in the `scripts/` directory. We believe this clarification will also help resolve the potential misunderstanding regarding our baselines, as discussed in Point 2.
>
> ---
> 2. **Add an Agentic Baseline**
>
> > Reviewer's Comments: "...the baselines are limited to single-turn, non-interactive prompting. This is a significant weakness...", "...please include at least one baseline that performs sequential, step-by-step reasoning and experimentation..."
>
> We respectfully disagree with the characterization of our baselines as "single-turn, non-interactive." Our baseline experiment implements a **fully interactive, sequential** experimentation framework. In each turn, the agent receives the **complete interaction history**—including all previous experimental inputs, observations, and hypothesis test results. Based on this accumulated evidence, the agent:
> 1. Updates its hypothesis
> 2. Designs new experiments
> 3. Decides whether to test the current hypothesis
>
> The Interface module will execute the proposed experiments and return the observations. If the agent chooses to test a hypothesis, the test result is returned. All interactions, i.e., experimental observations and test results, are incorporated into the historical context for subsequent turns, creating an iterative process.
>
> Our implementation can be viewed as a ReAct variant that preserves the core reasoning→acting loop while optimizing context usage. Instead of sequential turns (hypothesis → experiment → test), we batch operations within each turn (reason about all experiment history→ batch experiments + hypothesis + test decision). Table 1 in the paper demonstrates this interactivity, showing turn counts and hypothesis statistics.
>
> However, the reviewer’s comment about the README lacking "clear end-to-end instructions" may explain this misinterpretation. We will revise the README to include a guide for running the scripts for the full experiment, making the agentic and multi-turn nature of our framework unambiguous.
>
> ---
> 3. **Provide Task Generation Scripts**
>
> We thank the reviewer for this suggestion. We have included the task generation script and methodology details in our submission. Specifically:
> - `utils/preprocess.py`: Contains the conversion pipeline
> - `prompts/preprocess.txt`: Provides the prompts used for LLM-assisted conversion
>
> These resources enable the community to expand the benchmark and understand our methodology for converting problems from PHYBench or other resources into interactive discovery tasks.
>
> ---
> 4. **Quantify Verification Quality**
>
> We would like to clarify the details of our manual verification as follows.
> - Verification Team: 3 researchers with at least a master's degree, including one with an undergraduate background in physics.
> - LLM Pre-processing: We used Claude-3.7-sonnet, which achieved ~99% accuracy in initial conversion.
> - Manual Review Focus:
>   - Correct translation of formulas to Python format.
>   - Proper identification and listing of all variables.
>   - Accuracy of physical quantity names and descriptions relative to problem statements.
> - Quality Control: We removed several instances where answers were constants rather than formulas.
>
> ---
> 5. **Broader Impact**
>
> Thank you for the feedback. We will add a "Discussion" section to the paper, which will include remarks on limitations, future works, and broader impact.
>
> As a simulated benchmark, PHYSICIST has no direct societal impact, which is why we initially marked this as not applicable. However, we agree it is important to consider the downstream consequences of the research our work enables.
>
> Our work contributes positively by providing a safe, controlled environment to accelerate AI development while also offering novel tools to improve the transparency and reliability of AI reasoning. A key limitation is that the models we evaluate can inherit biases from training data, and our findings show that prior knowledge can affect them in complex, counterintuitive ways. PHYSICIST provides a platform to rigorously study these effects, contributing to the responsible development of more capable and robust AI systems.
>
> ---
> We thank the reviewer for their valuable feedback and hope that our responses have addressed their concerns comprehensively.

---

> > ### Comment · Reviewer_iUAt · 2025-08-04
> > **Reply for Rebuttal**
> >
> > Dear Authors,
> >
> > Thank you for your detailed rebuttal and for agreeing to several key improvements. We particularly appreciate the commitment to provide verification details and enhance the documentation. These changes will certainly strengthen the paper.
> >
> > My primary remaining concern is regarding the baseline. I understand your point about the turn-based interactivity. However, my core concern is that the baseline's reasoning process seems overly direct—**generating the entire plan for the next turn from a single prompt**. I think an agent that instead uses a more granular, multi-step reasoning process within a turn (e.g., a Thought/Action/Observation loop) would better showcase the interactive potential of PHYSICIST.
> >
> > In my opinion, the paper would be stronger if this design choice is discussed more explicitly, or if a more agentic baseline is considered. I would appreciate your thoughts on this point.
> >
> > Thank you again for the constructive dialogue.
> >
> > Sincerely,
> >
> > --Reviewer iUAt

---

> > > ### Author Response · Authors · 2025-08-05
> > >
> > > Dear Reviewer iUAt,
> > >
> > > Thank you for your constructive feedback and continued engagement with our work. We sincerely appreciate you taking the time to elaborate on your primary concern regarding the baseline, and we agree that a more comprehensive discussion of this design choice will strengthen the paper.
> > >
> > > Our current baseline provides a specific agent strategy: it retains a history of experimental observations, previously proposed hypotheses, and their test results. As shown in Table 1, this enables meaningful interactivity, as the agent can reference this history to formulate new hypotheses and strategically decide when to test them.
> > >
> > > While the agent generates its plan from a single prompt per turn, this design does not constrain it to overly direct reasoning. Instead, the complexity of the plan is flexible: the agent can choose to propose a single experiment (functionally equivalent to one step in a ReAct-style loop), or it can generate sophisticated multi-experiment plans with hypothesis formulation.
> > >
> > > For instance, the behavior of Claude-3.7-Sonnet on Env 39 (Level 3) illustrates this adaptive reasoning:
> > >
> > > - Turn 1 (Strategic Exploration): With no prior data, the agent recognizes it cannot form a hypothesis. It states its goal is to "design exploratory experiments to understand the relationship" and proceeds to generate a systematic batch of 10 experiments, explicitly choosing not to test a hypothesis at this stage.
> > > - Turn 2 (Preliminary Hypothesis & Targeted Testing): After receiving the initial data, the agent analyzes the results, identifies patterns (e.g., "T might be proportional to \sqrt{H}"), and forms a preliminary, incomplete hypothesis (2 * np.pi * np.sqrt(H / g)).  It then designs a new, targeted set of 5 experiments specifically to test and refine this initial idea.
> > > - Turn 4 (Analysis and Self-Correction): Finally, after a subsequent hypothesis is formally tested and fails, the agent re-evaluates the entire data history. Its reasoning shows clear self-correction, beginning with "The previous hypothesis ... was tested and found to be incorrect. Let me analyze the relationships more carefully...". It then correctly deduces the final physical relationships and formulates the correct formula.
> > >
> > > This example shows the LLM reasoning effectively in our interactive setting.
> > >
> > > Nevertheless, it is true that this approach differs from an agent that retains the entire granular reasoning history. This is a design choice driven by the challenge of managing the limited context window of LLMs, as the question of how to select or summarize this history is a non-trivial research problem in multi-turn agent design. Our baseline represents one practical solution.
> > >
> > > As you advised, our revision will include a detailed discussion of these agent design considerations. To better support the exploration of diverse agent strategies, we will enhance our ResearchInterface to facilitate the development and comparison of diverse architectures.
> > >
> > > Thank you again for your constructive comments.

---

> > > > ### Comment · Reviewer_iUAt · 2025-08-07
> > > > **Acknowledgement**
> > > >
> > > > Dear Authors,
> > > >
> > > > My apologies for the delay in this response.
> > > > Thank you for your detailed final rebuttal. Your explanations have sufficiently addressed my remaining concerns.
> > > >
> > > > The concrete example of the agent's reasoning process was informative for clarifying the capabilities of the current baseline. Additionally, the rationale provided for your design choice, particularly regarding the challenges of LLM context management, offered important context for your methodology.
> > > >
> > > > The commitments to discuss these design considerations in the revised paper and to enhance the framework for future research are valuable improvements to the work.
> > > >
> > > > Based on the resolution of these points, I have raised my score. Thank you for the constructive discussion.
> > > >
> > > > Sincerely,
> > > >
> > > > --Reviewer iUAt

---

> > > > > ### Author Response · Authors · 2025-08-07
> > > > >
> > > > > Dear Reviewer iUAt,
> > > > >
> > > > > Thank you for your reply. We are pleased to know that our response has addressed your concerns. We sincerely appreciate your continued engagement and the time you have dedicated to this process; your feedbacks have been invaluable in improving this work.

---

### Official Review · Reviewer_kgmZ · 2025-07-03

**Rating:** 5
**Confidence:** 3

**Summary:**

This paper introduces PHYSICIST, a test suite designed for benchmarking the scientific discovery capabilities of LLM-based agents within a simulated interactive environment. By controlling the amount of prior information providing to agents, PHYSICIST is able to test agents under various difficulty and obtain various results for comparison.

**Additional Feedback:**

1. It will be interesting to see how can the conclusions drawn from Physicist be used to build stronger scientific reasoning agents. For example, could we have a way to enhance the quality of the prior using the evaluated agent, given that prior is important for addressing Physicist tasks correctly.

**Dataset Code Accessibility:**

Yes

**Dataset Code Comments:**

N/A

**Ethical Considerations:**

No, there are no or only very minor ethics concerns

**Final Justification:**

Thanks the authors for the rebuttal, most of my concerns have been addressed clearly.

However, even though this is an initial study, conducting experiments such as fine-tuning would offer valuable insights into the challenge posed by the proposed benchmark. Specifically, if fine-tuning can easily solve the benchmark, it may indicate limitations in the benchmark's ability to reveal deeper deficiencies in LLM capabilities.

**Limitations Weaknesses:**

1. How can you control the task difficulty under extremely situations (e.g., level 4). For example, If most of the context/variables are masked, does the environment and the prior be able able to provide enough information for the agent to finish the task? Maybe some manual annotations can be added here.
2. It will be interesting to see whether LLMs can perform well on difficult tasks (level3/4) if they had been fine-tuned on such tasks. For example, the authors could try to create extra level3/4 tasks that are not overlapped with the original test set, and use these tasks to fine-tune an LLM. This fine-tuning experiments can help to understand whether LLMs could learn specific knowledge to solve such tasks during fine-tuning.

**Strengths Contributions:**

1, The proposed PHYSICIST can control the task difficulty easily, which extends the evaluation of scientific agents. Specifically, it has been meticulously designed and validated by human experts, and largely challenges SOTA LLMs, showing its effectiveness.
2. Experiments are designed and conducted comprehensively, which leads to several convincing findings such as the importance of prior selection when asking agents to perform scientific reasoning.

---

> ### Author Rebuttal · Authors · 2025-07-30
>
> We sincerely thank the reviewer for their positive evaluation and for recognizing the value of PHYSICIST in extending the evaluation of scientific agents. They proposed several constructive comments, which we would like to respond to as follows.
>
> ---
> **On Task Difficulty and Information Sufficiency at Extreme Levels**
>
> We thank the reviewer for this critical question about information sufficiency, which allows us to clarify the core philosophy of our benchmark. Our focus is less on theoretical solvability and more on how different levels of prior knowledge **practically affect an agent's scientific reasoning capabilities**.
>
> Determining a strict theoretical bound on solvability is difficult in such an interactive setting. Especially when considering physics priors, solvability is relative. Reconstructing Einstein's field equations purely from data would be a monumental and impossible task, yet for Einstein who with the right physical intuition, a few key observations suffice. Physics priors affect solvability in a sophisticated way. Regarding solvability with finite data, in general, arbitrary continuous functions cannot be determined from a finite number of points. However, even at Level 4, a powerful implicit prior remains: the agent knows it is analyzing a physics environment. This implies the underlying relationship is not an arbitrary function but an equation likely composed of low-order polynomials, power laws, or other simple forms. This implicit constraint reduces the effective search space.
>
> Theoretically, the tasks in Level 3 and Level 4 have the same underlying mathematical structure and thus the same formal solvability. However, our key finding is that the semantic priors from variable names in Level 3 provide a "namespace bias" that significantly improves practical performance over the anonymized Level 4. Moreover, the influence of priors is not always additive. We observe cases where a problem is solved at Level 3 but not at Level 2, despite Level 2 providing more explicit information. This highlights the complex, and sometimes counterintuitive, ways that agents interact with prior knowledge.
>
> Given these considerations, ultimately, the upper limit of a model's capability on these tasks is an open empirical question. This raises the prospect of exploring this performance ceiling through specialized training on Level 3 and Level 4 tasks, as the reviewer also insightfully suggests.
>
> ---
> **On Fine-Tuning LLMs for Difficult Tasks (Level 3/4)**
>
> This is an insightful suggestion. An experiment to fine-tune models on difficult tasks would indeed provide a much deeper understanding of whether the skills for low-prior scientific discovery can be explicitly learned.
>
> The work focusing on an experiment on Levels 3 and 4 can be positioned as a benchmark for interactive equation discovery. As we discuss in our Related Works section, such interactive benchmarks are currently lacking in the landscape, as most equation discovery methods are tailored to static datasets. Creating a large-scale, interactive benchmark for this purpose—perhaps by extending existing static benchmarks like LLM-SRBench —is a highly valuable goal for the community.
>
> However, the primary focus of our paper is to introduce the PHYSICIST framework and to analyze how an agent's performance changes across the spectrum of prior knowledge, from the context-rich reasoning required at Level 1 to the zero-prior discovery at Level 4. Therefore, while a dedicated fine-tuning experiment is a crucial next step, it is beyond the scope of this initial study.
>
> We will add this suggestion to our future work discussions. We believe that building on our framework to create a dedicated benchmark for interactive equation discovery and using it to fine-tune LLMs is a very important and promising research direction.
>
> ---
> **On Using Insights from PHYSICIST to Build Stronger Agents**
>
> We thank the reviewer for this forward-looking question. Our findings from PHYSICIST point to several concrete ways to build stronger scientific reasoning agents:
> - Improving Exploration Strategies: Our results show that current agents often employ naive or conservative sampling strategies. A more advanced agent could incorporate a dedicated exploration module trained to design experiments that maximize information gain or actively seek to falsify current hypotheses, mirroring real scientific practice more closely.
> - Automated Curriculum Learning: PHYSICIST's structured difficulty levels (L1-L4) provide a natural curriculum for training agents. An agent could be progressively trained, starting with full-context problems at Level 1 and gradually moving to the more challenging low-context scenarios at Level 4, thereby learning to rely more on interactive discovery.
> - Prior Enhancement: The reviewer's suggestion on prior construction represents a more distant but critical research direction. Our benchmark's multi-level setup allows us to analyze which specific pieces of prior knowledge are most crucial for solving a given task by comparing an agent's problem-solving process across different levels. This can help to develop a scientific reasoning agent that can efficiently query a knowledge source for useful information.
>
> We will include these discussions in the revised manuscript.
>
> ---
> We are confident that by incorporating these discussions, the revised manuscript will offer a more complete and impactful contribution to the community. We appreciate the reviewer's valuable guidance in strengthening our paper.

---

> > ### Comment · Reviewer_kgmZ · 2025-08-05
> >
> > Thanks the authors for the rebuttal, most of my concerns have been addressed clearly.
> >
> > However, even though this is an initial study, conducting experiments such as fine-tuning would offer valuable insights into the challenge posed by the proposed benchmark. Specifically, if fine-tuning can easily solve the benchmark, it may indicate limitations in the benchmark's ability to reveal deeper deficiencies in LLM capabilities.

---

> > > ### Author Response · Authors · 2025-08-06
> > >
> > > We thank the reviewer for their valuable feedback. We are glad that most of the reviewer's concerns have been addressed and we appreciate the opportunity to have further discussions.
> > >
> > > We agree that fine-tuning experiments represent an important extension of this work. For this study, our focus was to establish a zero-shot evaluation methodology, which is a common practice in other prominent LLM benchmarks like [BALROG](https://arxiv.org/pdf/2411.13543). This zero-shot approach is crucial for revealing the deficiencies in model capabilities that the reviewer mentioned, as it tests a model's generalization power without task-specific adaptation.
> > >
> > > Our experiments with leading proprietary models demonstrate the benchmark's significant challenge. The fact that these powerful LLMs struggled with our benchmark underscores its difficulty and confirms that the underlying challenges are non-trivial.
> > >
> > > As suggested, we will explicitly discuss fine-tuning as a critical avenue for future research in the revised manuscript.

---

### Official Review · Reviewer_MHHU · 2025-07-03

**Rating:** 5
**Confidence:** 3

**Summary:**

Authors introduce a novel benchmark suite and simulation platform for evaluating LLM-based scientific reasoning in interactive physics environments with fine-grained control over prior knowledge levels, and the development of tailored evaluation protocols that demonstrate the benchmark’s ability to differentiate model capabilities under varying conditions

**Dataset Code Accessibility:**

Yes

**Dataset Code Comments:**

Data is well-documented and usable with the given instructions

**Ethical Considerations:**

No, there are no or only very minor ethics concerns

**Final Justification:**

Rating: I maintain my 5 score
- Technically solid paper: Yes. Benchmark and experiments are well thought out and executed
- High impact on AI: Yes. Novel illustration of knowledge recall vs discovery in LLMs. Could be a key part of assessing whether future models can overcome these key challenges for novel knowledge discovery.

Confidence:
- Increased to 3 upon further review and discussion of the paper with the authors. However I don't see myself as a frontier expert in the paper's domain, so my confidence score is capped at 3, as I cannot eliminate the possibility of there being relevant work I am unfamiliar with, which could substantially change my evaluation of the paper if I were familiar with the literature.

**Limitations Weaknesses:**

Relatively minor comments, perhaps merely taste differences:
- I would strongly consider moving the "Case Studies" section from the appendix to the main body, as it's by far the strongest part of the paper to me. As (1) it contains what I consider the most novel and impactful findings of the paper, and (2) makes it much clearer why the "vary priors" setup is useful (i.e. I was initially underwhelmed by the results without seeing the actual examples, since I mistakenly interpreted the paper's headline result as merely "less context => less accuracy")
- Framing the main contribution of the paper as creating a new platform/framework/library seems to be underselling the paper's contributions greatly. I would much more prominently highlight the empirical demos of the conflict between LLM recall vs research (i.e. novel knowledge discovery) alongside the technical contribution, as it motivates the necessity of such an experimental framework much more strongly (since it yields surprising empirical results)
- If length is a concern, the Environment architecture/experiment implementation details seem less important to the paper than the results and case studies, so I would consider shortening to a brief high-level overview and save the details for the appendix
- I would be interested to explore the conflict between knowledge recall vs discovery even further by testing cases where the physical relationship to be discovered is not consistent with "real physics" (e.g. F=ma^2 instead of F=ma), which should very clearly highlight the limitations of models overly relying on facts memorized in their training data.

**Strengths Contributions:**

The authors discover many highly novel and illuminating examples that illustrate the limitations of LLMs for novel knowledge discovery, including:
- How models' "memorized knowledge" (or 'priors') affect their knowledge discovery ability by placing incorrect restrictions on their experimentation strategy (e.g. choosing only hypotheses that are well-represented in their training data, rather than "thinking outside the box" and exploring hypotheses that differ strongly from their priors but yields high information for their research)
- Tendency to overfit experimental data that is insufficient to deduce the true relationship (especially interesting given the novel context of deducing mathematical relationships, rather than standard ML predictions and regressions)
- Inconsistency between model's stated reasoning and their output/"final answer" demonstrating imperfect interpretability

To my knowledge this is the first instance in the literature of examples that clearly distinguish between knowledge discovery and knowledge recall, and experimental demonstration that these two LLM abilities can clash with each other, which is uniquely highlighted by the authors' experimental design allowing for varying priors

Suite seems comprehensive (covers many steps in the workflow for doing such AI experiments, convenient interface for tracking and running experiments) and should facilitate further experiments in this domain, including non-physics domain tests where such a masking experiment setup is desired (e.g. math, biology, chemistry, finance ...)

---

> ### Author Rebuttal · Authors · 2025-07-30
>
> We would like to thank the reviewer for their thorough, insightful, and positive assessment of our work. We appreciate that the reviewer recognized the core novelties of our findings and the potential of the PHYSICIST suite.
>
> Here is a detailed response to the comments:
>
> ---
> 1. **On Moving the "Case Studies" to the Main Body and Reframing the Main Contribution**
>
> We would like to thank the reviewer for their constructive suggestions. We agree that we should emphasize that the real value of PHYSICIST lies in the novel scientific insights it enables. We also believe this idea could be extended to many more scenarios besides physics. Our case studies indeed show some novel and surprising findings. They are essential for illustrating the complex, and sometimes counterintuitive, behaviors we observed, and also help with showing the necessity of a benchmark like PHYSICIST.
>
> In the revised manuscript, we will:
>
> - Move a condensed version of the "Case Studies" section and the Upset plot (Figure 9) from the appendix to the main body. It will validate that our experimental setup provides valuable insights into how models conduct scientific reasoning.
> - Revise the Introduction section to include more details about the empirical findings. We will emphasize that current LLMs exhibit surprising limitations in interactive discovery, a phenomenon we uncovered using PHYSICIST, the novel interactive benchmark we developed precisely to probe these capabilities.
>
> ---
> 2. **On Shortening Implementation Details**
>
> We agree with this assessment of content priorities. To accommodate case studies in the main body while adhering to length constraints, condensing the implementation details is indeed logical. We will shorten Section 3 (dataset schema and evaluation metrics) and Section 4 (model configurations), moving these details to the appendix.
>
> ---
> 3. **On Exploring "Non-Real Physics"**
>
> This is an insightful suggestion for future work. We are very excited about this idea.
>
> Your proposal to test relationships that are inconsistent with real-world physics gets to the heart of the challenge. Such an experiment would create a scenario where a model's reliance on memorized training data would become a clear disadvantage, forcing it to rely solely on its interactive, inductive reasoning capabilities. This is a powerful and direct way to explore the conflict between knowledge recall and discovery.
>
> There are some subtle points to consider. We need to create an environment that is internally consistent and physically plausible within its own stated rules, even if those rules contradict the physics of our universe. The units for all variables (mass, charge, length) should be correct, and the narrative of the context should remain plausible. Otherwise, the task may degenerate from a physics discovery problem into a mere symbolic regression or a math problem. An example could be that we modify some classical laws implicitly, which change the latent equation, but include only implicit hints in the context, like "novel material," "extreme temperatures," or "unusual geometric properties." In this way, the agent must rely on experiments to recover the underlying mechanism, which contradicts the known law.
>
> Our current Level 2 setting touches on this concept implicitly. For example, in Problem 310, when the context describing a mirror moving at relativistic velocity is masked, models fail to apply the necessary relativistic corrections. However, a dedicated suite of counterfactual physics problems would provide more direct evidence.
>
> This will be an important extension of our work. We will add this suggestion to our Discussion section. We believe this line of inquiry is crucial for developing AI scientists that can genuinely explore the unknown.
>
> ---
> We appreciate the detailed feedback from the reviewer. We are confident that incorporating these changes will strengthen our paper.

---

> > ### Comment · Reviewer_MHHU · 2025-08-05
> >
> > Thank you for your response. I look forward to the revised manuscript and perhaps future extension of your work

---

> > > ### Author Response · Authors · 2025-08-06
> > >
> > > Thank you for your feedback. We will incorporate all the promised updates in the revised manuscript. Thank you once again for dedicating your valuable time to helping us improve the quality of our paper.

---

### Official Review · Reviewer_8qZw · 2025-07-04

**Rating:** 5
**Confidence:** 4

**Summary:**

This paper introduces PHYSICIST, a benchmark and dataset designed to evaluate the scientific discovery capabilities of Large Language Models (LLMs). The authors address the rigidity of existing benchmarks by proposing a framework that allows for fine-grained control over the level of prior knowledge provided to an LLM agent. They also provide a dataset of different physics discovery tasks and show the effectiveness of this framework by providing performance metrics across three widely-used LLMs. This is a promising starting point to generate datasets and tasks for LLM agent based symbolic regression and broader automated scientific discovery.

**Dataset Code Accessibility:**

Yes

**Dataset Code Comments:**

The provided data file on dataverse has all the data as described in the paper. There are also LLM outputs in the file but it is not clear which specific models they are from.

I ran the code as described in the supplementary information.
After setting up the environment with the provided `requirements.txt`, I got an error on running the claude script.
`ModuleNotFoundError: No module named 'requests'`

I manually installed `requests`.
Please add requests explicitly to `requirements.txt`.

When running Claude test, another error popped up
`Could not find ANTHROPIC_API_KEY in environment variables or common .env files (including api_keys.env).`

Please state this in the readme. Right now it only asks you to add OpenRouter API key.
After adding my Anthropic key, the test ran smoothly.

Please make changes to the instructions to account for these errors.

**Ethical Considerations:**

No, there are no or only very minor ethics concerns

**Final Justification:**

The rebuttal resolves all practical concerns (missing requests dep, API-key docs) and clarifies key methodological points: it explains how the four knowledge levels deliberately expose any memorised physics the LLM already holds; details the two-stage equivalence check (SymPy first, Gemini fallback) and will add an explicit worked example; and expands results to include Deepseek-R1 and an open-weight model (Qwen2-32B), improving accessibility. These additions, plus a promised qualitative case-study section, remove my main reservations. Two issues remain: task-complexity metrics are still heuristic and the fallback LLM introduces some subjectivity, but they are clearly acknowledged and do not undermine the benchmark’s value. Given the strengthened manuscript, I raise my recommendation to 5.

**Limitations Weaknesses:**

I have a few questions/concerns:
1. How do you control for existing knowledge embedded in the weights of the trained model? A model could already have latex solutions for common physics problems in training data. Please comment if this potential dataset leakage/memorization is sufficiently addressed in the framework.
2. How robust is LLM-based Equivalence Assessment? It seems this could also produce false results if the LLM confabulates. How is this accounted for? Also would the nature of your test change depending on the model used for this assessment? Adding some more details and maybe an example about this metric in section 2.4 would strengthen the paper.
3. The provided code also has Deepseek R1 as another model that was used. Can you also comment on performance of this reasoning model? Also, please add results with an open-weight model as well for easier reproducibility and better accessibility.
4. The results section already has many quantitative results comparing different aspects of the suite. Showing examples of hypotheses generated by different models, especially for common failure modes, would provide deeper insight into the models' reasoning processes beyond aggregate performance metrics.
5. As noted by the authors, task complexity measurements can be improved. There is a rich body of work in symbolic regression literature.

Minor point:
- Question 16 on the NeurIPS paper checklist should be answered with 'Yes' since this work deals primarily with LLMs.

**Strengths Contributions:**

- This is a well-written paper and the authors describe a clear gap: existing benchmarks are too rigid in the amount of prior knowledge and information to LLM models. The new dataset has 97 physics discovery tasks which can be provided to agents with varying amounts of prior knowledge (context, variable descriptions and variable names). There are four levels of prior knowledge. This controlled knowledge framework is well-motivated and well-designed.
- The provided dataset is well-designed with detailed context and solutions. This is useful by itself as an initial suite of tests and as a template for future extensions across possibly other domains.
- The testing framework which allows agents to get quantitative metrics for hypothesis through on-the-fly python simulation is novel and provides good feedback for both the models on hypothesis quality and researchers on model quality. The constraints (compute budget, oracle use) and evaluation metrics are also reasonable.
- The provided results across Gemini, GPT-o4 and Claude demonstrate that this framework can be used to analyze the performance of different models across physics discovery tasks and can be used to meaningfully analyze the characteristics of different agents.

---

> ### Author Rebuttal · Authors · 2025-07-30
>
> We sincerely thank the reviewer for their thoughtful and positive evaluation of our work. We're delighted that they found the paper well-written, the framework well-motivated, and the contributions valuable for automated scientific discovery. We address each point raised as follows.
>
> ---
> 1. **On Controlling for Existing Knowledge in LLM Weights and Data Leakage**
>
> Thank you for this critical question, which addresses the central motivation for our work. Our benchmark is not designed to eliminate the influence of a model's pre-existing knowledge, but rather to **precisely study how it affects posterior exploration and discovery**.
>
> The "prior knowledge" an agent has before an experiment is a combination of its **internal knowledge** (from training) and the **external information** about the current environment which we provide in the prompt. The levels act as control over this external information, which controls how much an agent can rely on its internal knowledge versus posterior experimental data. At Level 1, the rich context allows the model to reason primarily from its existing knowledge. Conversely, at Level 4, the model must rely almost entirely on observations from the interactive experiments it conducts.
>
> The reviewer's concern about memorized LaTeX solutions is a good example of the phenomenon we measure. This type of memorization can be reflected in the performance difference between Level 3 (with standard variable names) and the anonymized Level 4, which directly quantifies the influence of knowledge about variable naming from the training set.
>
> The risk of data leakage or memorization is mitigated by the interactive nature of our benchmark. While a model might have memorized a static solution, it cannot have memorized the novel sequence of experiments and hypothesis refinements required to succeed in PHYSICIST. This format necessitates generative reasoning over simple retrieval.
>
> ---
> 2. **On the Robustness of LLM-based Equivalence Assessment**
>
> This is a good point regarding the robustness of our evaluation protocol. We will clarify our methodology, which uses LLM assessment as a controlled, secondary step to a more rigorous primary method.
>
> Our evaluation is a two-stage process designed to maximize accuracy while accounting for edge cases. Our primary method uses the SymPy library for a rigorous and deterministic check of mathematical equivalence between the agent's proposed formula and the ground-truth equation. Symbolic parsers can be brittle and sometimes produce false negatives when encountering syntactically different but semantically identical expressions (e.g., np.pi vs. sympy.pi). To compensate for this, we use a capable reasoning model (Gemini 2.5 Flash) as a flexible fallback to re-evaluate these specific cases. To ensure reliability, we require the assessment LLM to output a structured judgment, including a confidence score and a detailed explanation, which allows us to verify its reasoning.
>
> We will detail this two-stage methodology and provide a concrete example, as you suggested. Furthermore, to enhance rigor and minimize reliance on the LLM assessment, we will work on expanding the parsing rules of our symbolic module to handle more of these edge cases automatically.
>
> ---
> 3. **On Including Results from Other Models (Deepseek, Open-Weight)**
>
> Thank you for this suggestion. We initially experimented with Deepseek-R1 during our exploratory phase, which achieved a 45.56% success rate across 90 environments at Level 1, placing its performance between o4-mini-high and Claude 3.7 Sonnet. For the final manuscript, we focused on three widely-used models from distinct providers, including both reasoning-enabled ('thinking') and standard models, to present a clear comparative analysis.
>
> We agree that including open-weight models would enhance reproducibility and accessibility. We have therefore conducted new experiments with **Qwen2-32B** (a thinking model) on 2×80GB A100 GPUs, using the following settings:  `max_tokens=3000`, `max_model_len=32768`, and `temperature=0.2`.
>
> Our preliminary results show that Qwen2-32B achieved a 10.31% success rate at Level 1. Among successful cases, the model averaged only 1 test per solution and generated an average of 1.3 unique hypotheses, indicating limited ability to adjust hypotheses based on experimental feedback.
>
> We will include comprehensive results across all four levels for both Deepseek-R1 and Qwen2-32B in the revised manuscript.
>
> ---
> 4. **On Including Qualitative Examples and Failure Modes**
>
> We agree that qualitative examples are essential for providing deeper insight beyond aggregate metrics. In the revised manuscript, we will integrate a condensed version of our **Case Studies** section (currently in Appendix C) into the main body of the paper. This section analyzes specific success and failure modes, showing the actual hypotheses generated by different models and dissecting their reasoning processes. For example, we discuss how models are constrained by priors in Env. 716 and fail to design effective experiments under low-information settings in Env. 310. This addition will provide the deeper insight that the reviewer has called for.
>
> ---
> 5. **On Improving Task Complexity Measurements**
>
> We agree and acknowledge this as a limitation in the paper. In this work, we chose simple, interpretable heuristics like variable count as a simple approximation of complexity. We acknowledge that symbolic regression literature offers more sophisticated metrics. We will strengthen this point in our limitations discussion and cite relevant work from this field to guide future refinements of the benchmark.
>
> ---
> **Minor Point**
>
> > Reviewer's comment: Question 16 on the NeurIPS paper checklist should be answered with 'Yes' since this work deals primarily with LLMs.
>
> Thank you for the correction regarding the NeurIPS checklist. Our previous understanding of this question was incorrect. We will update the answer for Question 16 to 'Yes'.
>
> **Code and Data Comments**
>
> Thank you very much for testing and for flagging these important practical issues. We will fix these issues accordingly.
>
> ---
> We would like to thank the reviewer again and hope our response has fully addressed their concerns.

---

> > ### Comment · Reviewer_8qZw · 2025-08-03
> >
> > The rebuttal resolves all practical concerns (missing requests dep, API-key docs) and clarifies key methodological points: it explains how the four knowledge levels deliberately expose any memorized physics the LLM already holds; details the two-stage equivalence check (SymPy first, Gemini fallback) and will add an explicit worked example; and expands results to include Deepseek-R1 and an open-weight model (Qwen2-32B), improving accessibility. These additions, plus a promised qualitative case-study section, remove my main reservations. Two issues remain: task complexity metrics are still heuristic and the fallback LLM introduces some subjectivity, but they are clearly acknowledged and do not undermine the benchmark’s value. Given the strengthened manuscript, I raise my recommendation to 5.

---

> > > ### Author Response · Authors · 2025-08-06
> > >
> > > We would like to thank the reviewer for their valuable feedback. We are pleased to know that our rebuttal has been accepted, and we will incorporate all of the promised updates into the revised version. Thank you again for your time and contribution to our work.

---

### Comment · Area_Chair_vTJa · 2025-08-03

Thanks to the authors for the comprehensive rebuttal and thanks to 8qZw for responding -- all other reviewers, please have a look!

---

### Note · Authors · 2025-08-15

We would like to extend our sincerest gratitude to the Area Chair and all four reviewers for their exceptionally thorough, constructive, and insightful feedback. The detailed engagement and thoughtful discussions have been invaluable and will lead to a significantly stronger and more impactful manuscript.

We are delighted by the consensus that `PHYSICIST` addresses a "clear and significant gap" in the evaluation of AI for science through a "well-motivated and well-designed" framework. The reviewers highlighted that our core innovation, i.e. fine-grained control over prior knowledge, is a "powerful tool" for disentangling memorization from "true mechanistic inference and exploration." This innovation enabled what one reviewer described as the "first instance in the literature" to experimentally demonstrate the clash between an LLM's knowledge recall and its capacity for novel discovery, which is pivotal for advancing AI research. We are proud that our "meticulously designed" benchmark and "comprehensive" experiments were recognized as a "solid foundation for future research" and a "promising starting point" for this emerging field.

The review process has highlighted several key areas for improvement, and we are committed to addressing them comprehensively in our revised manuscript. In summary, the key revisions will include:
- Strengthened Narrative: Moving a condensed version of the Case Studies to the main body and revising the Introduction section to include more details about the empirical findings.
- Expanded Experiments: Adding results for open-weight models (Deepseek-R1, Qwen3-32B, and gpt-oss-20b), and clarifying our two-stage evaluation protocol (SymPy + LLM fallback).
- Improved Documentation: Featuring unified licensing statements, corrected metadata, and an improved README with instructions on running full experiments.
- Richer Discussion: Adding discussion on our baseline's interactive design, incorporating suggested future work (exploring counterfactual physics, fine-tuning, and building stronger scientific reasoning agents), and including a broader impacts section.

We are confident that these revisions will fully address all points raised and solidify the paper’s position as a foundational contribution to the community. Thank you once again for your time and invaluable guidance.

---

### Decision · Program_Chairs · 2025-09-18

**Decision:**

Accept (poster)

**Comment:**

The authors introduce PHYSICIST, a novel interactive benchmark for evaluating LLM-based scientific reasoning in physics discovery tasks. The benchmark's key innovation lies in its sophisticated control over prior knowledge levels, allowing researchers to systematically study how agents perform scientific discovery under varying information constraints. All reviewers recognize PHYSICIST as a well-designed and valuable contribution that addresses a significant gap in AI evaluation. The technical implementation is comprehensive, and good physics problems were selected, namely 97 physics problems adapted from PhyBench. The rebuttal period successfully resolved most reviewer concerns. The authors committed to expanding results with additional models, enhancing documentation, etc.

Overall, PHYSICIST addresses a genuine need in the scientific AI evaluation landscape by providing the first interactive physics discovery benchmark with systematic prior knowledge control. The empirical insights about LLM limitations in scientific reasoning—particularly how memorized knowledge can hinder genuine discovery—represent novel and important findings for the community.

===== FINAL UPDATE FROM DB Track PCs ====

The final decision for this paper has been taken by the program chairs after consultation with the SACs. All Senior Area Chairs have ranked papers according to the feedback from the AC during the review process. We decided to leave the original meta-review to reflect the opinion of the AC in light of the initial discussions with reviewers and SAC.